# FLOW-BASED AUTOMATIC NEURAL OPERATOR WITH HARD PHYSICAL CONSTRAINTS

## ABSTRACT

Simulating physical systems governed by partial differential equations (PDEs) is crucial across science and engineering. Recently, generative models—exemplified by Flow Matching—have emerged as a highly competitive approach due to their ability to effectively model high-dimensional solution distributions. However, these models often struggle to ensure physical consistency, frequently violating fundamental conservation laws or boundary conditions. In this work, we propose Physics-Manifold Flow Matching (PMFM), a novel generative framework for PDE simulation that directly addresses this challenge. PMFM introduces two key innovations. First, it enforces strict, hard physical constraints by restricting the entire generative trajectory to a physical manifold defined by analytical equations, while employing a Geometric Guidance Mechanism (GGM) to maintain high-fidelity solutions. Second, to handle complex multi-physics problems, we introduce an Adaptive Constraint Projection Framework that learns to dynamically select and parameterize the currently active physical laws. We validate PMFM on several challenging systems that are highly sensitive to physical constraints, and the results show that our framework is significantly superior to state-of-the-art physics-informed generative models in producing physically valid, long-term-stable simulations. The code can be found at `https://anonymous.4open.science/r/PMFM-F127`.

## 1 INTRODUCTION

Partial differential equations (PDEs) are widely used in many fields, including seismology, electromagnetics, fluid dynamics, and related areas. Traditionally, numerical methods have been the primary tools for solving PDEs.(LeVeque & Leveque, 1992) However, while ensuring model accuracy, such approaches often come with prohibitive computational costs.

With the rapid development of neural networks, deep learning has made remarkable progress in solving PDEs and has gradually become a research hotspot. In 2018, Raissi et al. proposed Physics-Informed Neural Networks (Raissi et al., 2019), which embed physical laws into the training process of neural networks by minimizing PDE residual losses. Another line of research, neural operators, aims to learn mappings from functions to functions(Li et al., 2020b). These methods go beyond the traditional pointwise regression paradigm by directly approximating infinite-dimensional mappings from input functions to solution functions. Representative works include DeepONet (Lu et al., 2021) and the Fourier Neural Operator (FNO) (Li et al., 2020a). DeepONet employs a branch-trunk architecture to capture complex relationships between functions with high-dimensional inputs and outputs, while FNO leverages Fourier transforms to represent functions in the frequency domain and uses convolutional operations to efficiently capture global interactions. Together,thes approaches provide new deep learning pathways for solving PDEs.

In recent years, generative models have also shown great potential in solving PDEs by learning the underlying distribution of solution functions. Representative works include Score-Based Generative Models (Song & Ermon, 2019), which learn the score function to reverse a predefined noise process, thereby generating PDE solutions from pure noise (Song et al., 2020b). Another rapidly developing approach is the Probability Flow ODE (Lipman et al., 2022), which employs Flow Matching techniques to learn a vector field that smoothly transports a simple Gaussian distribution into the distribution of PDE solutions. These generative approaches open up new perspectives and provide powerful

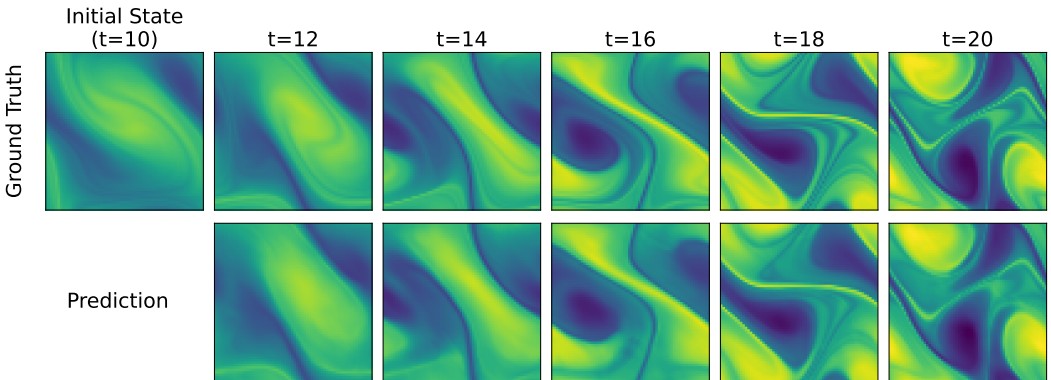

Figure 1: Comparison of ground truth (top) and PMFM prediction (bottom) for Navier-Stokes flow at Re=10,000

tools for addressing high-dimensional PDEs, performing uncertainty quantification, and exploring the solution space. Despite these advances, most existing deep learning methods rely heavily on black-box data-driven learning, often resulting in a lack of physical consistency. Specifically, such models may generate solutions that violate fundamental physical laws, such as mass conservation, energy conservation, or divergence-free conditions in fluid dynamics and electromagnetics (Raissi et al., 2019). These physically inconsistent outputs not only accumulate errors during long-term evolution, but also generalize poorly when extrapolated to unseen initial or boundary conditions (Wang et al., 2021). Furthermore, in the absence of inductive biases provided by physical laws, purely data-driven models typically require large amounts of training data and lack interpretability, limiting their reliability in scientific and engineering applications (Karniadakis et al., 2021).

To address these limitations in physical consistency and representational capacity, we propose a novel framework: Physics-Manifold Flow Matching (PMFM). Our method is built upon two key innovations:**(1) Generation on a guided physical manifold.** We enforce hard physical constraints by constraining the generative trajectory to a physical manifold, where all states are physically valid by definition. To preserve high-fidelity that are typically lost during the constraining projection, a Geometric Guidance Mechanism (GGM), which recycles the discarded residual information to guide the network's predictions and provably enhance model expressivity. **(2) Adaptive selection and parameterization of constraints.** To handle complex, multi-physics scenarios with dynamic constraints, we introduce an adaptive framework in which a neural gating network learns to dynamically select the relevant subset of constraints from an analytical library based on the current system state. Furthermore, this network can parameterize the constraints, allowing the model to learn unknown physical coefficients or adapt to state-dependent laws directly from data, thereby opening new up capabilities for system identification.

## 2 RELATED WORK

**Neural Operators.** Neural operators have been developed as infinite-dimensional generalizations of neural networks to approximate mappings between function spaces associated with PDEs (Lu et al., 2021; Li et al., 2020a; 2024). Representative architectures include the Fourier Neural Operator (FNO), which leverages spectral convolutions, and DeepONet, which learns operator mappings through a branch-trunk network design. Subsequent works have extended these frameworks to unstructured meshes (Boussif et al., 2022; Raonic et al., 2023), graph-based operators (Li et al., 2020b), multi-resolution and wavelet-based approaches (Gupta et al., 2021; Tripura & Chakraborty, 2023b), and attention-based designs (Hao et al., 2023; Wu et al., 2024) Recent developments also incorporate algebraic structures (Brandstetter et al., 2022) or Koopman theory (Xiong et al., 2024)to further enhance operator learning performance.

**Generative Models for PDEs.** Generative modeling has emerged as an alternative paradigm for learning PDE solution distributions. Score-based diffusion models have been applied to uncertainty

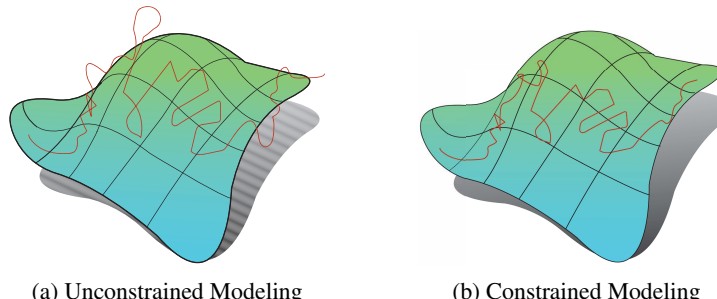

(a) Unconstrained Modeling          (b) Constrained Modeling

Figure 2: Comparison between (a) an unconstrained trajectory that deviates from the physical manifold and (b) our method's trajectory which is constrained to it.

quantification and PDE simulation (Song et al., 2020b; Song & Ermon, 2019; Dhariwal & Nichol, 2021; Hoogeboom et al., 2021; Huang et al., 2024). Flow-based approaches, including both discrete and continuous flow matching, further extend probabilistic modeling to high-dimensional spatiotemporal dynamics (Gat et al., 2024; Tong et al., 2023). Recent works have also introduced hard-constrained, projected, or constraint-aware diffusion and flow formulations (Christopher et al., 2024; Baldan et al., 2025; Cheng et al., 2025), illustrating growing interest in combining generative modeling with physical structure.

**Physics-Constrained Operator Learning.** Beyond purely data-driven approaches, many methods integrate physics directly into neural PDE frameworks. Classical techniques incorporate PDE residuals or physics-informed losses (Raissi et al., 2019). Recent works address conservation laws, boundary conditions, and constraint satisfaction through hard projections or constraint-guided learning (Hansen et al., 2023; Utkarsh et al., 2025b; Saad et al., 2023; Mouli et al., 2024). Projection-based constraint enforcement has also been explored for physical consistency (Chakraborty et al., 2020; Utkarsh et al., 2025a). Thermodynamics-inspired architectures further demonstrate complementary ways of embedding physical structure into learned models (Hernández et al., 2023).

## 3 PRELIMINARY

**Problem Definition.** We consider a class of partial differential equations (PDEs) defined on the domain $[0, T] \times D \subset \mathbb{R} \times \mathbb{R}^d$, in the form:

$$\frac{\partial u}{\partial t} = F\left(u, \frac{\partial u}{\partial x}, \frac{\partial^2 u}{\partial x^2}, \dots\right) + f(t, x), \quad (t, x) \in [0, T] \times D,$$

$$u(0, x) = u_0(x), \quad x \in D, \qquad B[u](t, x) = 0, \quad (t, x) \in [0, T] \times \partial D. \tag{1}$$

Here, $u : [0, T] \times D \to \mathbb{R}^n$ denotes the solution to the PDE, $u_0$ is the prescribed initial condition, and $B[\cdot]$ is the boundary operator. The term $F$ describes the governing dynamics, and $f(t, x)$ is an external forcing term.

For such PDE systems, in addition to approximating the solution $u$, we must also ensure the validity and physical consistency of the solution. Typical constraints include the divergence-free condition in incompressible fluid dynamics, Hamiltonian structures in conservative systems, and energy conservation laws in thermodynamic models.

**Flow Matching.** To approximate the solution distribution of PDEs, we adopt flow matching as the backbone generative model (Gat et al., 2024). Flow matching learns a continuous-time vector field $v_\theta(x, t)$ such that the trajectory $\{x_t\}_{t=0}^1$ evolves from a simple prior distribution $p_0$ (e.g., Gaussian noise) towards the target distribution $\pi$ of PDE solutions. Formally, the dynamics are defined as

$$\frac{dx_t}{dt} = v_\theta(x_t, t), \quad x_0 \sim p_0, \quad x_1 \sim \pi. \tag{2}$$

During training, the model is optimized to align $v_\theta$ with the ground-truth probability flow $v_t^\star$, by minimizing the flow matching loss:

$$\mathcal{L}_{\text{FM}} = \mathbb{E}_{x_0 \sim p_0, \, x_1 \sim \pi} \left[ \, \|v_\theta(x_t, t) - v_t^\star(x_t)\|^2 \, \right]. \tag{3}$$

This framework ensures that sampling can be performed by numerically integrating the learned ODE, starting from $x_0 \sim p_0$ and evolving along the trajectory $\{x_t\}_{t=0}^1$ until the final solution $x_1$. As a result, flow matching provides a flexible and efficient backbone for modeling PDE solution distributions.

## 4 OUR APPROACH

To precisely incorporate physical laws into generative modeling, we introduce the Physics-Manifold Flow Matching (PMFM) framework. Unlike traditional methods where trajectories evolve unconstrained in Euclidean space $\mathbb{R}^d$, PMFM forces the generative path to lie on a manifold defined by physical laws, guaranteeing physical consistency by construction. This is achieved through two core components: a projection mechanism that enforces hard constraints, and a geometric guidance mechanism that enhances model expressivity.

### 4.1 PHYSICS-MANIFOLD FLOW MATCHING

**Manifold-Constrained Dynamics.** At the heart of our framework is the physical manifold, $\mathcal{M}$, which represents the subspace of all physically valid states. It is defined as the set of points satisfying a series of constraint functions $C_i(u) = 0$, where each $C_i(u)$ embodies a physical law like mass or energy conservation:

$$\mathcal{M} := \{u \in \mathbb{R}^d : C_i(u) = 0, \, \forall i = 1, \ldots, m\} \tag{4}$$

For a trajectory $u_t$ to remain on $\mathcal{M}$, its velocity vector $\dot{u}_t$ must reside in the manifold's tangent space $T_u\mathcal{M}$ at every point. Formally, the tangent space is the kernel of the constraint Jacobian matrix $J_C(u)$:

$$T_u\mathcal{M} := \ker J_C(u), \quad \text{where} \quad J_C(u) = \begin{bmatrix} \nabla C_1(u)^\top & \cdots & \nabla C_m(u)^\top \end{bmatrix}^\top \in \mathbb{R}^{m \times n}. \tag{5}$$

In our method, a neural network first predicts an unconstrained vector field $v_\theta(u, t) \in \mathbb{R}^d$. To enforce the physical constraints, we orthogonally project this field onto the tangent space using the following operator.

**Definition 1 (Projection Operator).** *The projection operator $\Pi_{T_u\mathcal{M}} : \mathbb{R}^d \to T_u\mathcal{M}$ is defined as:*

$$\Pi_{T_u\mathcal{M}} := I - J_C(u)^\top \left(J_C(u)J_C(u)^\top\right)^{-1} J_C(u) \tag{6}$$

*where $I$ is the identity matrix. This assumes the constraint gradients are linearly independent, ensuring $J_C(u)J_C(u)^\top$ is invertible.*

The resulting physically-valid velocity field is $v_\mathcal{M}(u, t) = \Pi_{T_u\mathcal{M}} v_\theta(u, t)$. This formulation provides our core theoretical guarantee.

**Theorem 1 (Manifold Invariance).** *Given a physical manifold $\mathcal{M}$ and the tangential vector field $v_\mathcal{M}(u, t) = \Pi_{T_u\mathcal{M}} v_\theta(u, t)$, if an initial state $u_0 \in \mathcal{M}$, the trajectory governed by $\dot{u}_t = v_\mathcal{M}(u_t, t)$ remains on the manifold $\mathcal{M}$ for all time $t \geq 0$.*

**Geometric Guidance Mechanism for Enhanced Expressivity.** While ensuring physical feasibility, the projection operation discards the vector component orthogonal to the manifold. This "residual" component, however, may contain valuable information for modeling fine-grained details. To address this information loss, we introduce the Geometric Guidance Mechanism (GGM). For any unconstrained vector field $\tilde{v}(u, t)$ (e.g., a target field during training), the discarded residual $\mathbf{r}$ is:

$$\mathbf{r}(u, t) := \tilde{v}(u, t) - \Pi_{T_u\mathcal{M}}\tilde{v}(u, t) = (I - \Pi_{T_u\mathcal{M}})\tilde{v}(u, t) \tag{7}$$

The GGM encodes this residual into a latent variable $z = E_\phi(\mathbf{r})$ and uses it to guide the tangential dynamics *before* projection, ensuring constraints are never violated:

$$v_\mathcal{M}(u, t) := \Pi_{T_u\mathcal{M}}(g_\theta(u, t) + B_\theta(u, t)\alpha_\phi(z)) \tag{8}$$

Here, $g_\theta$ is a base vector field, the columns of $B_\theta$ form a basis for a subspace of the tangent space, and $\alpha_\phi(z)$ provides control coefficients learned from the residual information. This design strictly enhances model expressivity without compromising physical consistency.

**Theorem 2 (Consistency and Expressivity of GGM).** *The compensated dynamics defined by GGM are both consistent and expressive. If a target vector field is already tangential, the residual vanishes ($z = 0$) and the model degenerates to the standard form $v_\mathcal{M}(u,t) = \Pi_{T_u\mathcal{M}}\, g_\theta(u,t)$. For any sufficiently smooth target tangent vector field on $\mathcal{M}$, there exist network parameters that allow the compensated dynamics to approximate it arbitrarily well.*

Finally, the PMFM framework extends beyond single trajectories. By guaranteeing the velocity field $v_\mathcal{M}$ is always tangential, PMFM ensures the evolution of an entire probability density $\rho_t(u)$ correctly satisfies the manifold continuity equation, $\partial_t \rho_t + \mathrm{div}_\mathcal{M}(\rho_t v_\mathcal{M}) = 0$. This inherently enforces physical consistency at the distributional level.

## 4.2 Training and Sampling with Geometric Guidance

**Learning with Geometric Guidance.** We use the Conditional Flow Matching (CFM) objective to train the network parameters $\Theta = \{\theta, \phi\}$. The core idea is to have the network's predicted compensated vector field, $v_{\text{comp}}$, approximate a target conditional vector field, $v_{\text{target}}(u_t, t) = u_1 - u_0$, which points from a prior sample $u_0 \sim p_0(u)$ to a real data sample $u_1 \sim p_{\text{data}}(u)$. To effectively train the Geometric Guidance Mechanism (GGM), we first compute the residual component (i.e., the normal component) of this target field:

$$\mathbf{r}_t = (I - \Pi_{T_{u_t}\mathcal{M}})(u_1 - u_0) \tag{9}$$

This residual is then transformed by the encoder $E_\phi$ into a latent variable $z_t = E_\phi(\mathbf{r}_t)$ used for conditional guidance. Subsequently, the network predicts the compensated field based on the current state $u_t$ and the geometric guidance $z_t$:

$$v_{\text{comp}}(u_t, t, z_t; \Theta) = g_\theta(u_t, t) + B_\theta(u_t, t)\alpha_\phi(z_t) \tag{10}$$

We optimize the network parameters $\Theta$ by minimizing the mean squared error between the predicted field and the target field. The loss function is defined as follows:

$$\mathcal{L}(\Theta) = \mathbb{E}_{t, u_0, u_1} \left\| v_{\text{comp}}(u_t, t, z_t; \Theta) - (u_1 - u_0) \right\|^2 \tag{11}$$

Crucially, since physical consistency is structurally guaranteed by the projection operator during the inference phase, our loss function requires no additional penalty terms or complex regularization. The complete training procedure is detailed in Algorithm 1.

**Physically-Compliant Data Generation.** After the network is trained, we can sample $u_0$ from a simple prior distribution (e.g., a Gaussian distribution) and generate complex, physically-compliant data samples by numerically solving the following ordinary differential equation (ODE):

$$\frac{du_t}{dt} = v_\mathcal{M}(u_t, t) = \Pi_{T_{u_t}\mathcal{M}} \left( v_{\text{comp}}(u_t, t, z_t; \Theta) \right) \tag{12}$$

The dynamics here are driven by our trained network $v_{\text{comp}}$. The key point is that during inference, no target field exists, and thus there is no residual information. Accordingly, we set the latent variable for geometric guidance to the zero vector, i.e., $z_t = \mathbf{0}$. Finally, after the last step of the ODE solver, we perform an additional projection operation, $u_1 \to \Pi_{T_{u_1}\mathcal{M}} u_1$, to correct for any minor cumulative errors from numerical integration, ensuring the final output strictly satisfies all physical constraints.

## 4.3 Adaptive Constraint Projection Framework

In many real-world complex physical systems, the constraints themselves may be dynamic or activated only in specific regions and under certain conditions. To address this complexity and further enhance the model's flexibility and expressivity, we design an adaptive constraint projection framework. The core idea of this framework is to enable the model to dynamically construct and apply constraints in a data-driven manner based on the current state, rather than relying on a fixed, globally active set of constraints.

**Library of Parameterizable Constraints**    To strike a balance between the rigor of hard constraints and the complexity of dynamic systems, we first design a library of adaptive analytical constraints $\mathcal{C} = \{C_i(u; \alpha_i)\}_{i=1}^K$. The essence of this design is that each constraint $C_i$ in the library has two aspects: a fixed analytical structure and a variable adaptive parameter. Its analytical structure, such as $C(u; \alpha) = u|_{\partial\Omega} - \alpha$ for a boundary condition, is a non-learnable mathematical form determined by physical priors. This ensures that we can compute its exact analytical gradient at any time, providing a solid mathematical foundation for constructing the hard constraint projection operator. Within this structure, we reserve one or more parameters $\alpha_i \in \mathbb{R}^{p_i}$ as tuning knobs, whose values are predicted in real-time by a neural network based on the system state, thereby endowing the constraint itself with the ability to adapt dynamically.

**Gating and Parameterization Network**    To intelligently schedule the constraint library, we design a neural network $G_\psi$ (with parameters $\psi$) to act as the framework's control module. At each time step $t$, for the current system state $u_t$, the network $G_\psi(u_t, t)$ simultaneously performs two tasks. First, it outputs a set of logits $l_t \in \mathbb{R}^K$ to perform gating, which involves determining the set of active constraint indices $S_t = \{i \mid \sigma(l_{t,i}) > 0.5\}$ through a thresholding mechanism. Concurrently, the network completes the parameterization task by predicting the parameters $A_t = \{\alpha_i(t)\}_{i=1}^K$ for all potential constraints, but in subsequent computations, we only use the parameters corresponding to the active set $S_t$.

**Dynamic Constraint Composition and Projection.**    The true innovation of this framework lies in its ability to dynamically construct an instantaneous projection operator at each time step based on the real-time output of the gating network. This process follows a rigorous mathematical procedure. First, using the active set $S_t$ and the corresponding parameters $\alpha_i(t)$, we define the instantaneous physical manifold $\mathcal{M}_t$ at the current state:

$$\mathcal{M}_t := \{u \in \mathbb{R}^d \mid C_i(u; \alpha_i(t)) = 0, \forall i \in S_t\} \tag{13}$$

Next, by stacking the gradients of all active constraints, we construct the dynamic Jacobian matrix $J_{S_t}(u_t)$ corresponding to this instantaneous manifold:

$$J_{S_t}(u_t) := \left[\nabla_u C_{i_1}(u_t; \alpha_{i_1}(t))^\top; \ldots; \nabla_u C_{i_{|S_t|}}(u_t; \alpha_{i_{|S_t|}}(t))^\top\right], \quad i_1, \ldots, i_{|S_t|} \in S_t. \tag{14}$$

Finally, using this dynamic Jacobian, we generate a dynamic projection operator $\Pi_{T_{u_t}\mathcal{M}_t}$ to enforce all currently active constraints:

$$\Pi_{T_{u_t}\mathcal{M}_t} := I - J_{S_t}(u_t)^\top \left(J_{S_t}(u_t) J_{S_t}(u_t)^\top\right)^{-1} J_{S_t}(u_t) \tag{15}$$

Ultimately, the velocity vector that ensures the trajectory evolves on the dynamically changing manifold $\mathcal{M}_t$ is given by:

$$\dot{u}_t = v_{\mathcal{M}_t}(u_t, t) = \Pi_{T_{u_t}\mathcal{M}_t}\left(v_{\text{comp}}(u_t, t, z_t; \Theta)\right) \tag{16}$$

Through this adaptive framework, our model not only guarantees strict physical hard constraints but also automatically learns how to intelligently select, parameterize, and combine multiple physical laws during complex spatio-temporal evolution, greatly enhancing its capability and potential for tackling real-world problems.

## 5 EXPERIMENTS

In this section, we aim to demonstrate **(1)** the ability of PMFM to generate high-fidelity and physically consistent solutions for a variety of complex, nonlinear partial differential equations (PDEs) **(2)** its effectiveness in long-term, probabilistic forecasting, where our method is expected to maintain stability and accuracy over extended temporal rollouts **(3)** its superior generative performance compared to state-of-the-art baselines.

### 5.1 EXPERIMENTAL SETUP

**Datasets.**    All datasets we use come from PDEBench (Takamoto et al., 2022), including 1D time-dependent Advection equation, 1D Burgers' equation, 2D Darcy Flow, 2D Advection equation and a challenging 2D Navier-Stokes problem with velocity forcing. Each dataset is split into 1000/100/100 samples for training, validation, and testing.

**Baselines.** We benchmark our PMFM framework against several state-of-the-art baselines: (1) the foundational DeepONet (Lu et al., 2021) (2) the widely-adopted Fourier Neural Operator (FNO) (Li et al., 2020a) (3) the Wavelet Neural Operator (WNO) (Tripura & Chakraborty, 2023a) for its superior gradient capture capability, and (4) the Latent Spectral Model (LSM) (Wu et al., 2023). To specifically evaluate our structure-preserving framework's contribution, we additionally compare against DDPM (Ho et al., 2020) enhanced with DDIM sampling acceleration (Song et al., 2020a), isolating the impact of our physical constraints from the base generative model's performance. Complete implementation details and hyperparameter configurations are provided in Appendix D.

Table 1: Comparison of our proposed PMFM model against baseline methods across PDE benchmark datasets. We report the average L2 relative error(L2 Rel) and Mean Squared Error (MSE). For clarity, the best results are highlighted in **bold**, and the second-best results are underlined.

| | Dataset | | | | | | | | | |
| | 1D Burgers' | | 1D Advection | | 2D Darcy Flow | | 2D Advection | | 2D Navier-Stokes | |
| Metric | L2 Rel | MSE | L2 Rel | MSE | L2 Rel | MSE | L2 Rel | MSE | L2 Rel | MSE |
|---|---|---|---|---|---|---|---|---|---|---|
| DeepONet | 3.08e-01± 6.10e-03 | 4.11e-02± 9.57e-04 | 2.50e-02± 3.10e-03 | 1.04e-03± 1.34e-04 | 1.67e-01± 3.50e-03 | 2.33e-04± 4.82e-05 | 8.20e-03± 3.00e-04 | 9.67e-05± 6.43e-07 | 3.64e-01± 2.30e-03 | 2.89e-01± 4.71e-03 |
| FNO | 6.98e-02± 0.00e+00 | 5.03e-03± 3.82e-05 | 2.30e-02± 4.00e-04 | 3.93e-04± 1.59e-05 | 9.02e-02± 7.00e-04 | 1.17e-04± 2.76e-06 | 7.80e-03± 2.00e-04 | 7.16e-05± 9.12e-07 | 1.84e-01± 6.00e-04 | 7.93e-02± 5.47e-04 |
| WNO | 1.04e-01± 4.80e-03 | 1.23e-02± 1.90e-03 | 2.23e-02± 9.00e-04 | 4.08e-04± 8.97e-06 | 8.97e-02± 5.10e-03 | 1.69e-04± 5.28e-06 | 1.85e-02± 1.00e-04 | 3.97e-04± 1.61e-06 | 2.38e-01± 2.60e-03 | 1.28e-01± 2.82e-03 |
| LSM | 1.98e-01± 2.54e-02 | 4.20e-02± 8.59e-05 | 9.76e-01± 2.76e-02 | 1.28e+00± 7.26e-02 | **7.31e-02±** **4.00e-04** | **7.02e-05±** **9.47e-06** | 2.28e-02± 1.30e-03 | 2.45e-04± 4.82e-05 | 2.12e-01± 1.70e-03 | 1.13e-01± 5.23e-03 |
| DDPM | 9.63e-02± 2.30e-03 | 7.85e-03± 3.92e-04 | 1.99e-02± 7.00e-04 | 2.62e-04± 4.27e-05 | 1.23e-01± 1.10e-03 | 1.92e-04± 3.66e-05 | 1.54e-02± 4.00e-04 | 8.33e-05± 5.08e-06 | 2.78e-01± 3.20e-03 | 1.76e-01± 9.48e-03 |
| D-Flow | 7.83e-02± 3.50e-03 | 3.86e-03± 1.32e-04 | 1.16e-02± 2.00e-03 | 6.88e-05± 4.27e-06 | 1.35e-01± 4.10e-03 | 1.97e-04± 8.91e-06 | 1.16e-02± 9.00e-04 | 6.87e-05± 3.08e-06 | 2.20e-01± 5.00e-03 | 1.51e-01± 2.10e-02 |
| PMFM (ours) | **5.44e-02±** **2.00e-04** | **2.81e-03±** **2.27e-04** | **5.60e-03±** **5.00e-04** | **4.68e-05±** **7.71e-06** | 7.71e-02± 2.20e-03 | 1.01e-04± 1.47e-05 | **6.00e-03±** **3.00e-04** | **5.62e-05±** **3.33e-06** | **1.46e-01±** **8.37e-03** | **4.62e-02±** **9.56e-3** |

**Implementation details.** All experiments are based on PyTorch and trained on a single RTX 5090 GPU.To reduce computational cost during training, we apply appropriate downsampling to the original high-resolution dataset.For all experiments, we employed the AdamW optimizer with a learning rate of $1 \times 10^{-3}$ and a weight decay of $1 \times 10^{-5}$. To evaluate the accuracy of the generated solutions, we use two primary metrics: the Mean Squared Error (MSE), defined as $\frac{1}{N} \sum_{i=1}^{N} (u_i - \hat{u}_i)^2$, and the relative $L_2$ error, $\frac{\|u - \hat{u}\|_2}{\|u\|_2}$, where $u$ denotes the ground-truth solution and $\hat{u}$ is the model's prediction over all grid points.

## 5.2 MAIN RESULTS AND DISCUSSION

Benchmark results are reported in Table 1. We can see that our model achieves SOTA performance on four benchmarks (Burgers, Advec1d, Advec2d and NS2D, reducing relative $L_2$ error by 22%, 74%, 23% and 15% respectively, validating the exceptional performance of our PMFM framework.

Remarkably, our method excels in temporal predictions, notably for the Advec1d dataset that describes conserved-quantity transport where both mass and shape should remain invariant. Traditional numerical schemes are prone to introduce numerical dissipation or dispersion errors during long-term integration, leading to the smearing of sharp features and the decay of the total mass, thereby violating the core conservation law. The success of PMFM is mainly attributed to the core theorem of the PMFM framework: Manifold Invariance. Using a projection operator to constrain the vector field to the tangent space ($T_{u_t}\mathcal{M}$) of the physical manifold ($\mathcal{M}$) at every step of evolution, our method ensures that the trajectory remains within the space of physically feasible solutions. This fundamentally suppresses the long-standing problem of cumulative numerical dissipation found in traditional methods, thereby enabling ultra-high-fidelity long-term predictions.

For Burgers' equation with challenging shock discontinuities, PMFM achieves state-of-the-art performance. Conventional generative models that rely on soft constraints tend to prioritize capturing smooth, low-frequency signal components, often struggling to generate sharp discontinuities. However, PMFM perfectly solves this issue through its unique residual-aware compensation mechanism. Specifically, the high-frequency, non-smooth information contained in the shock front is captured primarily in the orthogonal residual component, which is discarded during projection onto

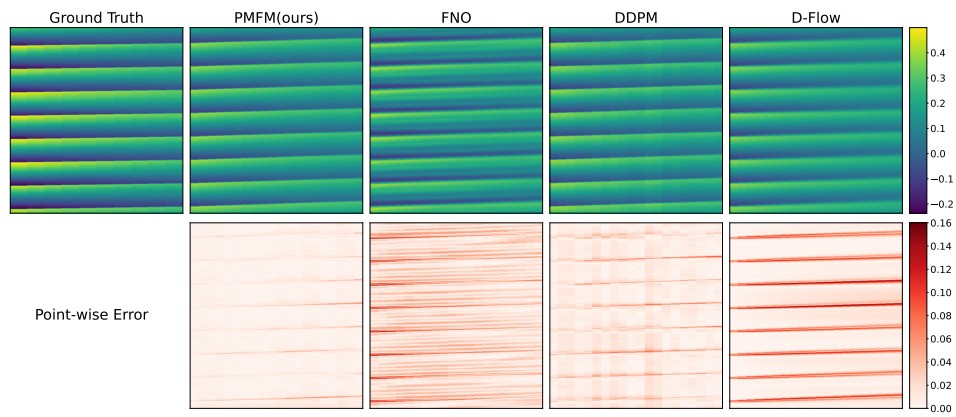

Figure 3: Comparative Visualization of PMFM and Baseline Models on Burgers' Equation.

the smooth tangent space. Our model innovatively encodes this residual into a latent variable $z$ and injects its information back into the tangential dynamics. This allows PMFM to reconstruct the sharp features of the shock front with high fidelity while strictly guaranteeing manifold invariance.

In the Darcy flow problem, our model achieves the second-best performance with a slight margin compared to the SOTA model (LSM). However, in comparison to other models based on the same generative paradigm, our PMFM model demonstrates a significant reduction in error. Traditional generative models like flow matching permit unrestricted Euclidean-space trajectories, failing to ensure physical consistency in generated solutions and often producing deviant results with prediction errors. In contrast, the "hard constraint" design of PMFM fundamentally ensures that every point along the evolution path, from the initial state to the final solution, is fully physically feasible, thereby avoiding the physical inconsistency errors. For more complex real-world problems, our designed Adaptive Constraint Projection Framework has even greater potential to dynamically select and combine physical constraints.

### 5.3 LONG-TERM PREDICTION STABILITY.

As shown in Figure 5, PMFM exhibits significantly slower error growth compared to baseline methods throughout the entire prediction horizon. Notably, in the early stages, PMFM effectively suppresses rapid error accumulation , demonstrating its inherent stability. The progressive divergence in prediction accuracy confirms PMFM's superior capability in long-term dynamical system modeling, as its physics-informed architecture inherently mitigates error accumulation mechanisms that plague conventional approaches.

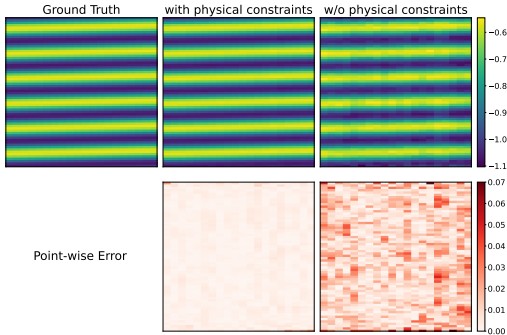

Figure 4: 1D advection at $t = 20$: PMFM (left) vs. no-physics ablation (right).

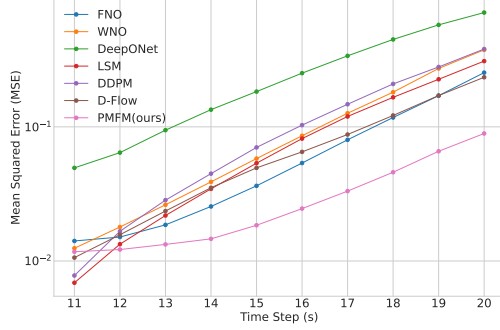

Figure 5: Performance comparison on Navier-Stokes equations ($Re = 10000$) over increasing prediction horizons (11-20 timesteps).

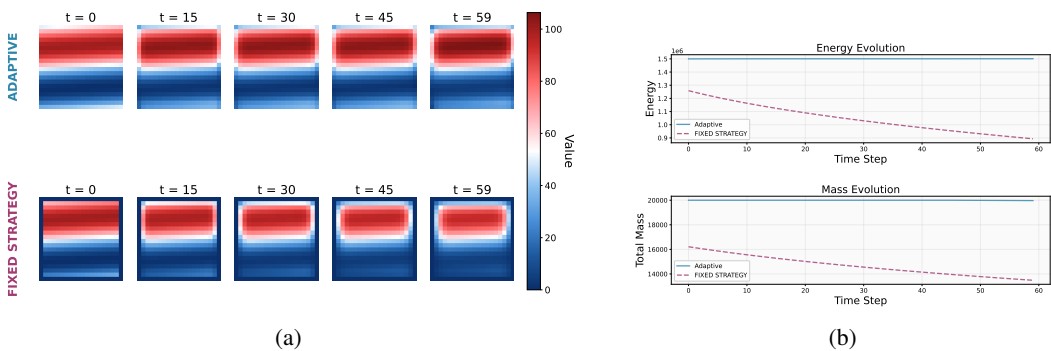

(a)                                                                                          (b)

Figure 6: Comparative analysis between the adaptive framework and fixed strategies: (a) System state snapshots at critical time points; (b) Evolution of energy and mass metrics throughout the process.

## 5.4    ABLATION STUDY

**Physical constraints.**    To quantitatively evaluate the importance of the core physical constraint mechanism in our Physics-Manifold Flow Matching (PMFM) framework, we compare our full PMFM model against an ablated variant, denoted w/o Physical constraints, The results of our ablation study are presented in Table 2, with a detailed error profile visualized in Figure 4.Remarkably, the error of the w/o Physical constraints model is not only higher on average but also exhibits progressively blurred deviations in the later stages of the sequence. This performance drop occurs because the unconstrained model systematically violates conservation laws due to lacking physical guidance. This results in predictions with drifts in conserved quantities (e.g., mass and energy), leading to physically distorted trajectories. In contrast, by enforcing solutions to remain on the physical manifold, our PMFM model structurally guarantees both physical realism and predictive accuracy.

**Adaptive Constraint Projection Framework.**    To demonstrate the effectiveness and necessity of our adaptive mechanism, we conducted an ablation study comparing our full Adaptive Constraint Projection Framework against a simplified baseline using a fixed, non-adaptive constraint strategy. This baseline, referred to as FIXED STRATEGY, employs a conventional approach with static, pre-defined constraints—in this case, fixed zero-value boundary conditions—without dynamically enforcing any conservation laws.Results are reported in Figure 6.The fixed strategy causes rapid decay of the system state, with continuous energy and mass loss, resembling unconstrained diffusion. In contrast, our adaptive framework dynamically enforces conservation laws, maintaining stable energy and mass, ensuring physically plausible simulations. This confirms that adaptive constraints are essential for accuracy, while fixed strategies fail to preserve physical integrity.

## 6    CONCLUSION

We introduce Physics-Manifold Flow Matching (PMFM), a novel generative framework that structurally guarantees physical consistency by constraining the entire generative process to the physical manifold. By integrating tangent-space projection with a Geometrically-Guided Mechanism (GGM), our method excels at producing long-term, stable, and physically valid simulations, especially for systems containing sharp features such as shocks. Its ability to recover high-fidelity details while strictly obeying hard constraints marks a significant advance over traditional physics-informed models. Moreover, the proposed adaptive constrained-projection module provides flexibility for handling dynamic constraints. A key direction for future work is to extend this framework to automatically discover unknown physical laws from data, eliminating the current reliance on pre-defined constraint libraries and ultimately yielding more trustworthy generative models for science and engineering.

**LLM Usage**    LLMs were utilized to assist with the linguistic polishing of this manuscript. All scientific concepts, methodologies, and analyses were developed exclusively by the authors.

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

APPENDIX

# A  THEORETICAL PROOFS

## A.1  PROOF OF THEOREM 1 (MANIFOLD INVARIANCE)

*Proof.* Let the physical manifold be $\mathcal{M} = \{u \in \mathbb{R}^d : C_i(u) = 0, \ i = 1, \ldots, m\}$ with Jacobian $J_C(u) = [\nabla C_1(u)^\top; \ldots; \nabla C_m(u)^\top]$. By Definition 1, the tangent space at $u \in \mathcal{M}$ is $T_u\mathcal{M} = \ker J_C(u)$. Consider the projected (tangential) dynamics

$$\dot{u}_t = v_\mathcal{M}(u_t, t) = \Pi_{T_{u_t}\mathcal{M}} v_\theta(u_t, t), \tag{17}$$

where $\Pi_{T_u\mathcal{M}}$ is the orthogonal projector onto $T_u\mathcal{M}$ given in Definition 2. Assume $u_0 \in \mathcal{M}$ and that $C$ is $C^1$ in a neighborhood of $\mathcal{M}$. For each constraint $C_i$, by the chain rule,

$$\frac{\mathrm{d}}{\mathrm{d}t} C_i(u_t) = \nabla C_i(u_t)^\top \dot{u}_t = \left(J_C(u_t) \Pi_{T_{u_t}\mathcal{M}} v_\theta(u_t, t)\right)_i. \tag{18}$$

Since $\mathrm{range}(\Pi_{T_u\mathcal{M}}) = T_u\mathcal{M} = \ker J_C(u)$, we have $J_C(u_t) \Pi_{T_{u_t}\mathcal{M}} = 0$ for all $t$ such that $u_t \in \mathcal{M}$. Hence $\frac{\mathrm{d}}{\mathrm{d}t} C_i(u_t) = 0$ for all $i$. Because $C_i(u_0) = 0$, it follows that $C_i(u_t) \equiv 0$ for all $t \geq 0$, i.e., $u_t \in \mathcal{M}$ for all $t \geq 0$. This proves manifold invariance. $\square$

## A.2  PROOF OF THEOREM 2 (CONSISTENCY AND EXPRESSIVITY OF COMPENSATED DYNAMICS)

*Proof.* **Assumptions.** We work on a compact set $\mathcal{K} \subset \mathcal{M}$ and assume: (i) the projector $\Pi_{T_u\mathcal{M}}$ is well-defined and smooth in $u \in \mathcal{K}$ (full row rank of $J_C(u)$); (ii) the encoder and the coefficient head satisfy the natural nulling property $E_\phi(0) = 0$ and $\alpha_\phi(0) = 0$; (iii) $g_\theta$, $B_\theta$ and $\alpha_\phi$ are universal approximators on $\mathcal{K}$ (e.g., feedforward neural networks with non-polynomial activations); (iv) for each $u \in \mathcal{K}$, the columns of $B_\theta(u, t)$ span (or can approximate an arbitrary basis of) a subspace dense in $T_u\mathcal{M}$.[1]

**Consistency.** Suppose the target field $v^*(u, t) \in T_u\mathcal{M}$ for all $(u, t) \in \mathcal{K} \times [0, 1]$. Then, by definition of the residual,

$$\mathbf{r}(u, t) = (I - \Pi_{T_u\mathcal{M}}) v^*(u, t) = 0, \tag{19}$$

hence $z = E_\phi(\mathbf{r}) = E_\phi(0) = 0$ and $\alpha_\phi(z) = \alpha_\phi(0) = 0$ by (ii). Therefore the compensated field reduces to

$$v_\mathcal{M}(u, t) = \Pi_{T_u\mathcal{M}}\big(g_\theta(u, t) + B_\theta(u, t)\alpha_\phi(z)\big) = \Pi_{T_u\mathcal{M}}g_\theta(u, t). \tag{20}$$

If, in addition, $g_\theta$ is trained (or chosen) so that $\Pi_{T_u\mathcal{M}}g_\theta(u, t) = v^*(u, t)$, the compensated dynamics coincide with the standard physics-constrained form. This proves the consistency part.

**Expressivity.** Let $v_{\text{target}}(u, t) \in T_u\mathcal{M}$ be any sufficiently smooth tangent field on $\mathcal{K} \times [0, 1]$ and $\varepsilon > 0$. Because $\Pi_{T_u\mathcal{M}}$ acts as the identity on $T_u\mathcal{M}$, it suffices to construct $\tilde{v}(u, t)$ such that $\|\Pi_{T_u\mathcal{M}}\tilde{v}(u, t) - v_{\text{target}}(u, t)\| < \varepsilon$ uniformly. There are two equivalent constructions.

*(a) Base-field realization.* By (iii) there exists $g_\theta$ with $\|g_\theta(u, t) - v_{\text{target}}(u, t)\| < \varepsilon$ on $\mathcal{K} \times [0, 1]$. Applying the projector and using its continuity (assumption (i)) and the fact that $v_{\text{target}}(u, t) \in T_u\mathcal{M}$,

$$\big\|\Pi_{T_u\mathcal{M}}g_\theta(u, t) - v_{\text{target}}(u, t)\big\| = \big\|\Pi_{T_u\mathcal{M}}(g_\theta(u, t) - v_{\text{target}}(u, t))\big\| \leq \|\Pi_{T_u\mathcal{M}}\| \varepsilon, \tag{21}$$

which can be made $< \varepsilon$ by rescaling $\varepsilon$. Setting $z \equiv 0$ (hence $\alpha_\phi(z) \equiv 0$) yields $v_\mathcal{M}(u, t) = \Pi_{T_u\mathcal{M}}g_\theta(u, t)$ approximating $v_{\text{target}}$ arbitrarily well.

*(b) Affine control realization.* Alternatively, by (iv) we can choose $B_\theta(u, t)$ to approximate a local basis of $T_u\mathcal{M}$. Then there exists a coefficient map $\beta(u, t)$ such that $B_\theta(u, t)\beta(u, t)$ approximates $v_{\text{target}}(u, t)$ uniformly. By (iii) we can approximate $\beta(u, t)$ with $\alpha_\phi(z(u, t))$ for some latent code $z(u, t)$, and again

$$v_\mathcal{M}(u, t) = \Pi_{T_u\mathcal{M}}\big(g_\theta(u, t) + B_\theta(u, t)\alpha_\phi(z(u, t))\big) \approx v_{\text{target}}(u, t). \tag{22}$$

---

[1] A sufficient condition is that $B_\theta(u, t)$ can approximate a moving frame of $T_u\mathcal{M}$; see, e.g., standard universal approximation arguments on compacta.

Either construction shows that the model class is dense in the space of continuous tangent fields on $\mathcal{M}$ over $\mathcal{K} \times [0, 1]$. Hence the compensated dynamics possess the stated expressivity.

Combining the two parts concludes the proof of simultaneous consistency and expressivity. $\qquad\square$

**Remarks on regularity and numerics.** (i) The proofs assume $C_i$ are $C^1$ and that $J_C(u)J_C(u)^\top$ is full-rank on $\mathcal{M}$ so that the projector is well-defined and smooth. (ii) In practice, one uses a numerically stable pseudoinverse (e.g., with Tikhonov regularization) to realize $\Pi_{T_u\mathcal{M}}$ without affecting the invariance argument.

# B  SPECIFIC FORMULATIONS OF PHYSICAL CONSTRAINT OPERATORS

In our Physics-Manifold Flow Matching (PMFM) framework, all physical constraints are imposed by defining a physical manifold $\mathcal{M}$ through a set of constraint functions $C_i(\mathbf{u}) = 0$. The core mechanism is the projection operator $\Pi_{T_{\mathbf{u}}\mathcal{M}}$, which maps any vector onto the tangent space of the manifold. This ensures that the evolution velocity remains within the tangent space, thereby keeping the trajectory on the manifold.

The general form of this projection operator is given by:

$$\Pi_{T_{\mathbf{u}}\mathcal{M}} := \mathbf{I} - \mathbf{J}_C(\mathbf{u})^\top (\mathbf{J}_C(\mathbf{u})\mathbf{J}_C(\mathbf{u})^\top)^{-1}\mathbf{J}_C(\mathbf{u}) \tag{23}$$

where $\mathbf{J}_C(\mathbf{u})$ is the Jacobian matrix of all active constraint functions.

In this section, we derive the specific forms of this operator for several key physical properties.

## B.1  GEOMETRIC / KINEMATIC CONSTRAINTS

### B.1.1  INCOMPRESSIBILITY (DIVERGENCE-FREE)

**Physical Meaning**  For an incompressible fluid, the velocity field $\mathbf{u}$ must be divergence-free at every point.

**Constraint Function** $C(\mathbf{u})$  For a 2D velocity field $\mathbf{u} = (u_x, u_y)$ discretized on a staggered grid, the divergence at cell $(i, j)$ can be approximated using finite differences:

$$C_{i,j}(\mathbf{u}) = (\nabla \cdot \mathbf{u})_{i,j} = \frac{u_{x,i+1,j} - u_{x,i,j}}{\Delta x} + \frac{u_{y,i,j+1} - u_{y,i,j}}{\Delta y} = 0 \tag{24}$$

The full constraint vector $C(\mathbf{u})$ is formed by stacking the constraints $C_{i,j}(\mathbf{u})$ for all cells in the domain.

**Jacobian Matrix** $\mathbf{J}_C(\mathbf{u})$  Each row of the Jacobian corresponds to the gradient $\nabla C_{i,j}(\mathbf{u})$ with respect to the entire flattened velocity vector. Since each $C_{i,j}$ only depends on four neighboring velocity components, the Jacobian is a large, sparse matrix. This matrix is, in effect, the discrete **divergence operator**, denoted by $\mathbf{D}$. Thus, we can write:

$$\mathbf{J}_C(\mathbf{u}) = \mathbf{D} \tag{25}$$

**Projection Operator** $\Pi_{T_{\mathbf{u}}\mathcal{M}}$  Substituting $\mathbf{J}_C(\mathbf{u}) = \mathbf{D}$ into the general formula, we obtain the projection operator for incompressibility:

$$\Pi_{\text{div-free}} = \mathbf{I} - \mathbf{D}^\top (\mathbf{D}\mathbf{D}^\top)^{-1}\mathbf{D} \tag{26}$$

This operator projects an arbitrary vector field onto its divergence-free component, which is a cornerstone of the Helmholtz-Hodge decomposition.

## B.2  CONSERVATION LAWS

### B.2.1  MASS / VOLUME CONSERVATION

**Physical Meaning**  The total mass (or volume, for an incompressible fluid) of the system remains constant over time.

**Constraint Function** $C(\mathbf{u})$   Let $\mathbf{u}$ be a vector representing the density in each cell. The total mass is the sum over all cells. For a conserved mass $M_0$ and assuming uniform cell volumes $V_i = 1$, the constraint is:

$$C(\mathbf{u}) = \sum_{i=1}^{N} u_i - M_0 = 0 \tag{27}$$

**Jacobian Matrix** $\mathbf{J}_C(\mathbf{u})$   The gradient of this single constraint function is a row vector of ones:

$$\mathbf{J}_C(\mathbf{u}) = \nabla C(\mathbf{u})^\top = [1, 1, \ldots, 1] = \mathbf{1}^\top \tag{28}$$

**Projection Operator** $\Pi_{T_\mathbf{u}\mathcal{M}}$   We substitute this into the general formula. Since $\mathbf{J}_C(\mathbf{u})\mathbf{J}_C(\mathbf{u})^\top = \mathbf{1}^\top \mathbf{1} = N$ (where $N$ is the total number of cells), we get:

$$\Pi_{\text{mass-consv}} = \mathbf{I} - \frac{1}{N}\mathbf{1}\mathbf{1}^\top \tag{29}$$

This operator subtracts the mean from any update vector, ensuring that the sum of the vector's components is zero and thus the total mass is conserved.

### B.2.2   ENERGY CONSERVATION

**Physical Meaning**   The total energy of the system, such as kinetic energy, remains constant.

**Constraint Function** $C(\mathbf{u})$   For kinetic energy, assuming unit mass per cell, the total energy is $E = \frac{1}{2}\sum_i u_i^2$. The constraint is:

$$C(\mathbf{u}) = \frac{1}{2}\sum_{i=1}^{N} u_i^2 - E_0 = \frac{1}{2}\mathbf{u}^\top \mathbf{u} - E_0 = 0 \tag{30}$$

**Jacobian Matrix** $\mathbf{J}_C(\mathbf{u})$   The gradient is simply the state vector $\mathbf{u}$ itself:

$$\mathbf{J}_C(\mathbf{u}) = \nabla C(\mathbf{u})^\top = [u_1, u_2, \ldots, u_N] = \mathbf{u}^\top \tag{31}$$

**Projection Operator** $\Pi_{T_\mathbf{u}\mathcal{M}}$   Substituting into the general formula, with $\mathbf{J}_C(\mathbf{u})\mathbf{J}_C(\mathbf{u})^\top = \mathbf{u}^\top \mathbf{u} = ||\mathbf{u}||_2^2$:

$$\Pi_{\text{energy-consv}} = \mathbf{I} - \frac{\mathbf{u}\mathbf{u}^\top}{||\mathbf{u}||_2^2} \tag{32}$$

This operator projects any update vector onto the space orthogonal to the current state vector $\mathbf{u}$. This ensures that any change is orthogonal to $\mathbf{u}$, which geometrically means the length of $\mathbf{u}$ remains constant, thereby conserving kinetic energy.

## B.3   BOUNDARY CONDITIONS

### B.3.1   DIRICHLET BOUNDARY CONDITIONS

**Physical Meaning**   The value of the solution is fixed at the domain boundaries $\partial D$. A common example is the no-slip condition where velocity is zero.

**Constraint Function** $C(\mathbf{u})$   We can define a selection matrix $\mathbf{S}_b$ that extracts the boundary components from the state vector $\mathbf{u}$. The constraint is:

$$C(\mathbf{u}) = \mathbf{S}_b\mathbf{u} - \mathbf{u}_{\text{boundary}} = 0 \tag{33}$$

where $\mathbf{u}_{\text{boundary}}$ contains the prescribed constant values at the boundary.

**Jacobian Matrix** $\mathbf{J}_C(\mathbf{u})$   The Jacobian is simply the selection matrix itself:

$$\mathbf{J}_C(\mathbf{u}) = \mathbf{S}_b \tag{34}$$

**Projection Operator** $\Pi_{T_\mathbf{u}\mathcal{M}}$    Substituting into the formula gives:

$$\Pi_{\text{Dirichlet}} = \mathbf{I} - \mathbf{S}_b^\top (\mathbf{S}_b \mathbf{S}_b^\top)^{-1} \mathbf{S}_b \tag{35}$$

Since each row of $\mathbf{S}_b$ selects a unique component, the rows are orthogonal, making $\mathbf{S}_b \mathbf{S}_b^\top = \mathbf{I}$. The operator simplifies to:

$$\Pi_{\text{Dirichlet}} = \mathbf{I} - \mathbf{S}_b^\top \mathbf{S}_b \tag{36}$$

The matrix $\mathbf{S}_b^\top \mathbf{S}_b$ is a diagonal matrix with ones at indices corresponding to boundary nodes and zeros elsewhere. Therefore, this operator effectively zeroes out the components of any update vector that correspond to the boundary nodes, ensuring their values remain fixed throughout the evolution.

### B.4    GEOMETRIC / KINEMATIC CONSTRAINTS

#### B.4.1    PERIODIC BOUNDARY CONDITIONS

**Physical Meaning**    The field values (and selected derivatives) are identical on opposite faces of a periodic domain.

**Constraint Function** $C(\mathbf{u})$    Let $\mathbf{S}_{\text{per}}$ encode pairwise equality between periodic dof indices (each row enforces $u_{i_\text{L}} - u_{i_\text{R}} = 0$). Then

$$C(\mathbf{u}) = \mathbf{S}_{\text{per}}\mathbf{u} = \mathbf{0}. \tag{37}$$

**Jacobian Matrix** $\mathbf{J}_C(\mathbf{u})$    Each row contains a $+1$ at $i_\text{L}$ and a $-1$ at $i_\text{R}$:

$$\mathbf{J}_C(\mathbf{u}) = \mathbf{S}_{\text{per}}. \tag{38}$$

**Projection Operator** $\Pi_{T_\mathbf{u}\mathcal{M}}$

$$\Pi_{\text{periodic}} = \mathbf{I} - \mathbf{S}_{\text{per}}^\top (\mathbf{S}_{\text{per}} \mathbf{S}_{\text{per}}^\top)^{-1} \mathbf{S}_{\text{per}}. \tag{39}$$

#### B.4.2    NO-SLIP (VELOCITY) DIRICHLET ON SOLID WALLS

**Physical Meaning**    Velocity is zero on impermeable solid boundaries.

**Constraint Function** $C(\mathbf{u})$    With $\mathbf{S}_b$ selecting boundary velocity components,

$$C(\mathbf{u}) = \mathbf{S}_b\mathbf{u} - \mathbf{0} = \mathbf{0}. \tag{40}$$

**Jacobian Matrix** $\mathbf{J}_C(\mathbf{u})$

$$\mathbf{J}_C(\mathbf{u}) = \mathbf{S}_b. \tag{41}$$

**Projection Operator** $\Pi_{T_\mathbf{u}\mathcal{M}}$

$$\Pi_{\text{no-slip}} = \mathbf{I} - \mathbf{S}_b^\top (\mathbf{S}_b \mathbf{S}_b^\top)^{-1} \mathbf{S}_b = \mathbf{I} - \mathbf{S}_b^\top \mathbf{S}_b. \tag{42}$$

#### B.4.3    NEUMANN (FLUX) BOUNDARY CONDITIONS

**Physical Meaning**    The normal derivative (or flux) at the boundary is prescribed.

**Constraint Function** $C(\mathbf{u})$    Let $\partial_n$ be a discrete normal derivative operator on $\partial D$, and $\mathbf{q}$ the prescribed boundary flux values. With $\mathbf{S}_\partial$ selecting boundary nodes,

$$C(\mathbf{u}) = \mathbf{S}_\partial (\partial_n \mathbf{u}) - \mathbf{q} = \mathbf{0}. \tag{43}$$

**Jacobian Matrix** $\mathbf{J}_C(\mathbf{u})$

$$\mathbf{J}_C(\mathbf{u}) = \mathbf{S}_\partial \, \partial_n. \tag{44}$$

**Projection Operator** $\Pi_{T_\mathbf{u}\mathcal{M}}$

$$\Pi_{\text{Neumann}} = \mathbf{I} - (\partial_n^\top \mathbf{S}_\partial^\top)(\mathbf{S}_\partial \partial_n \partial_n^\top \mathbf{S}_\partial^\top)^{-1} \mathbf{S}_\partial \partial_n. \tag{45}$$

### B.4.4 ROBIN (MIXED) BOUNDARY CONDITIONS

**Physical Meaning** A linear combination of value and normal derivative is prescribed on $\partial D$.

**Constraint Function $C(\mathbf{u})$** With scalars (or diagonal operators) $a, b$ and target $r$ on boundary dof,

$$C(\mathbf{u}) = \mathbf{S}_\partial \big(a\,\mathbf{u} + b\,\partial_n\mathbf{u}\big) - r = \mathbf{0}. \tag{46}$$

**Jacobian Matrix $\mathbf{J}_C(\mathbf{u})$**

$$\mathbf{J}_C(\mathbf{u}) = \mathbf{S}_\partial\big(a\,\mathbf{I} + b\,\partial_n\big). \tag{47}$$

**Projection Operator $\Pi_{T_\mathbf{u}\mathcal{M}}$**

$$\Pi_{\text{Robin}} = \mathbf{I} - \big(a\,\mathbf{I} + b\,\partial_n^\top\big)\mathbf{S}_\partial^\top\Big(\mathbf{S}_\partial(a\,\mathbf{I} + b\,\partial_n)(a\,\mathbf{I} + b\,\partial_n)^\top\mathbf{S}_\partial^\top\Big)^{-1}\mathbf{S}_\partial(a\,\mathbf{I} + b\,\partial_n). \tag{48}$$

### B.4.5 ZERO NET FLUX / ZERO MEAN (GAUGE) CONDITIONS

**Physical Meaning** Remove null-space modes by fixing zero mean (e.g., pressure) or zero net flux.

**Constraint Function $C(\mathbf{u})$** For zero mean:

$$C(\mathbf{u}) = \frac{1}{N}\mathbf{1}^\top\mathbf{u} = 0. \tag{49}$$

**Jacobian Matrix $\mathbf{J}_C(\mathbf{u})$**

$$\mathbf{J}_C(\mathbf{u}) = \frac{1}{N}\mathbf{1}^\top. \tag{50}$$

**Projection Operator $\Pi_{T_\mathbf{u}\mathcal{M}}$**

$$\Pi_{\text{zero-mean}} = \mathbf{I} - \frac{1}{N}\mathbf{1}\mathbf{1}^\top. \tag{51}$$

## B.5 CONSERVATION LAWS

### B.5.1 FIRST MOMENT (CENTER-OF-MASS) EVOLUTION IN 1D ADVECTION

**Physical Meaning** For constant advection speed $c$ on a periodic domain, the first spatial moment (center-of-mass weighted by $u$) shifts linearly in time.

**Constraint Function $C(\mathbf{u})$** Let $x_j$ be grid coordinates and $\Delta x$ the spacing. With prescribed $M_1(t) = M_1(0) + c\,t\,M_0$ and $M_0 = \Delta x\sum_j u_j$,

$$C(\mathbf{u}; t) = \Delta x\sum_j x_j u_j - M_1(t) = 0. \tag{52}$$

**Jacobian Matrix $\mathbf{J}_C(\mathbf{u})$**

$$\mathbf{J}_C(\mathbf{u}) = \Delta x\,[x_1, x_2, \ldots, x_N]. \tag{53}$$

**Projection Operator $\Pi_{T_\mathbf{u}\mathcal{M}}$**

$$\Pi_{\text{COM-advect}} = \mathbf{I} - \frac{\big(\Delta x\,\mathbf{x}\big)\big(\Delta x\,\mathbf{x}\big)^\top}{\|\Delta x\,\mathbf{x}\|_2^2}. \tag{54}$$

### B.5.2 FOURIER MAGNITUDE INVARIANCE (PURE TRANSLATION)

**Physical Meaning** Pure advection preserves the amplitude spectrum of the scalar field; only phases drift linearly in time.

**Constraint Function** $C(\mathbf{u})$   Let $\widehat{\mathbf{u}} = \mathbf{Fu}$ be the DFT, and $\alpha_k = |\widehat{u}_k(0)|^2$. For selected low/mid bands $\mathcal{K}$,

$$C_k(\mathbf{u}) = |\widehat{u}_k|^2 - \alpha_k = 0, \quad k \in \mathcal{K}. \tag{55}$$

**Jacobian Matrix** $\mathbf{J}_C(\mathbf{u})$   Using $\widehat{\mathbf{u}} = \mathbf{Fu}$ and Wirtinger calculus,

$$\frac{\partial |\widehat{u}_k|^2}{\partial \mathbf{u}} = 2\,\Re\{\widehat{u}_k^* \, \mathbf{F}_{k,:}\}, \quad \Rightarrow \quad \mathbf{J}_C(\mathbf{u}) = \begin{bmatrix} 2\,\Re\{\widehat{u}_{k_1}^* \, \mathbf{F}_{k_1,:}\} \\ \vdots \\ 2\,\Re\{\widehat{u}_{k_m}^* \, \mathbf{F}_{k_m,:}\} \end{bmatrix}. \tag{56}$$

**Projection Operator** $\Pi_{T_\mathbf{u}\mathcal{M}}$

$$\Pi_{\text{spec-mag}} = \mathbf{I} - \mathbf{J}_C^\top \big(\mathbf{J}_C \mathbf{J}_C^\top\big)^{-1} \mathbf{J}_C. \tag{57}$$

### B.5.3   GLOBAL FLUX BALANCE (DARCY)

**Physical Meaning**   Net boundary flux equals the integral of volumetric sources/sinks.

**Constraint Function** $C(\mathbf{p})$   With pressure $\mathbf{p}$, permeability $K$, and source $f$,

$$C(\mathbf{p}) = \int_{\partial\Omega} K\nabla p \cdot n \, ds - \int_\Omega f \, dx = 0. \tag{58}$$

A discrete quadrature yields $C(\mathbf{p}) = \mathbf{w}_\partial^\top \mathbf{G} \mathbf{p} - \mathbf{w}_\Omega^\top \mathbf{f}$ where $\mathbf{G}$ is a boundary-gradient operator and $\mathbf{w}_\partial, \mathbf{w}_\Omega$ are weights.

**Jacobian Matrix** $\mathbf{J}_C(\mathbf{p})$

$$\mathbf{J}_C(\mathbf{p}) = \mathbf{w}_\partial^\top \mathbf{G}. \tag{59}$$

**Projection Operator** $\Pi_{T_\mathbf{p}\mathcal{M}}$

$$\Pi_{\text{flux-bal}} = \mathbf{I} - \frac{(\mathbf{G}^\top \mathbf{w}_\partial)(\mathbf{w}_\partial^\top \mathbf{G})}{\|\mathbf{w}_\partial^\top \mathbf{G}\|_2^2}. \tag{60}$$

### B.6   PDE RESIDUAL CONSTRAINTS

### B.6.1   DARCY COLLOCATION RESIDUAL

**Physical Meaning**   The Darcy equation holds pointwise in the domain.

**Constraint Function** $C(\mathbf{p})$   On collocation set $\{x_m\}_{m=1}^M$ with discrete operator $\mathbf{L}$ approximating $\nabla \cdot (K\nabla)$,

$$C(\mathbf{p}) = \mathbf{Lp} - \mathbf{f} = \mathbf{0}, \qquad \text{(rows indexed by collocation points)}. \tag{61}$$

**Jacobian Matrix** $\mathbf{J}_C(\mathbf{p})$

$$\mathbf{J}_C(\mathbf{p}) = \mathbf{L}. \tag{62}$$

**Projection Operator** $\Pi_{T_\mathbf{p}\mathcal{M}}$

$$\Pi_{\text{Darcy-res}} = \mathbf{I} - \mathbf{L}^\top (\mathbf{L}\mathbf{L}^\top)^{-1} \mathbf{L}. \tag{63}$$

### B.7   FLUID-SPECIFIC INTEGRAL / STRUCTURAL CONSTRAINTS

### B.7.1   INCOMPRESSIBILITY (DIVERGENCE-FREE) ON A COLLOCATED GRID (VARIANT)

**Physical Meaning**   Same as the staggered-grid case; a collocated stencil differs in discrete coefficients.

**Constraint Function** $C(\mathbf{u})$    For 2D velocity $\mathbf{u} = (u, v)$ on a collocated grid,

$$C_{i,j}(\mathbf{u}) = \frac{u_{i+1,j} - u_{i-1,j}}{2\Delta x} + \frac{v_{i,j+1} - v_{i,j-1}}{2\Delta y} = 0. \tag{64}$$

**Jacobian Matrix** $\mathbf{J}_C(\mathbf{u})$    Stack the divergence stencil rows into a sparse matrix $\mathbf{D}_{\mathrm{coll}}$:

$$\mathbf{J}_C(\mathbf{u}) = \mathbf{D}_{\mathrm{coll}}. \tag{65}$$

**Projection Operator** $\Pi_{T_{\mathbf{u}}\mathcal{M}}$

$$\Pi_{\mathrm{div\text{-}free(collocated)}} = \mathbf{I} - \mathbf{D}_{\mathrm{coll}}^{\top}(\mathbf{D}_{\mathrm{coll}}\mathbf{D}_{\mathrm{coll}}^{\top})^{-1}\mathbf{D}_{\mathrm{coll}}. \tag{66}$$

### B.7.2 ENERGY BALANCE (HARDIFIED EQUALITY, TIME-DISCRETE)

**Physical Meaning**    For viscous flows, kinetic energy decays due to dissipation and may gain from external forcing; we enforce a discrete balance as an equality using an auxiliary nonnegative slack.

**Constraint Function** $C(\mathbf{u}^{n+1}, \zeta)$    With time step $\Delta t$, viscosity $\nu$, midpoint state $\mathbf{u}^{n+\frac{1}{2}}$, and forcing work $W^{n+\frac{1}{2}}$,

$$C(\mathbf{u}^{n+1}, \zeta) = \tfrac{1}{2}\|\mathbf{u}^{n+1}\|_2^2 - \tfrac{1}{2}\|\mathbf{u}^n\|_2^2 + \Delta t \cdot \nu\|\nabla\mathbf{u}^{n+\frac{1}{2}}\|_2^2 - \Delta t \cdot W^{n+\frac{1}{2}} - \zeta^2 = 0, \tag{67}$$

where $\zeta \geq 0$ converts the inequality into an equality in an augmented state.

**Jacobian Matrix** $\mathbf{J}_C(\mathbf{u}^{n+1}, \zeta)$

$$\mathbf{J}_C = \big[\, (\mathbf{u}^{n+1})^{\top} \quad \partial_{\mathbf{u}^{n+1}}\big(\Delta t \cdot \nu\|\nabla\mathbf{u}^{n+\frac{1}{2}}\|_2^2 - \Delta t \cdot W^{n+\frac{1}{2}}\big) \quad -2\zeta \,\big], \tag{68}$$

where the middle block is a linear operator from the discrete gradient/forcing at the chosen time discretization.

**Projection Operator** $\Pi_{T_{\mathbf{z}}\mathcal{M}}$    On the augmented state $\mathbf{z} = [\mathbf{u}^{n+1}; \zeta]$,

$$\Pi_{\mathrm{energy\text{-}bal}} = \mathbf{I} - \mathbf{J}_C^{\top}(\mathbf{J}_C\mathbf{J}_C^{\top})^{-1}\mathbf{J}_C. \tag{69}$$

## C  DERIVATIONS AND NUMERICAL PROJECTION

### C.1  GENERAL ORTHOGONAL PROJECTION ONTO CONSTRAINT TANGENT SPACE

Let $u \in \mathbb{R}^d$ satisfy equality constraints $C(u) = 0$ with Jacobian $J_C(u) \in \mathbb{R}^{m \times d}$. Given an unconstrained increment $v \in \mathbb{R}^d$, we seek the nearest increment $v^{\star}$ that lies in the tangent space $\mathcal{T}_u\mathcal{M} = \ker J_C(u)$:

$$\min_{\delta v} \|\delta v\|_2^2 \quad \text{s.t.} \quad J_C(u)\,(v + \delta v) = 0. \tag{70}$$

The Lagrangian $L(\delta v, \lambda) = \|\delta v\|_2^2 + \lambda^{\top} J_C(u)(v + \delta v)$ yields

$$\delta v^{\star} = -J_C(u)^{\top}\big(J_C(u)J_C(u)^{\top}\big)^{-1}J_C(u)\,v, \tag{71}$$

hence the orthogonal projector onto $\mathcal{T}_u\mathcal{M}$ is

$$\Pi(u) = I - J_C(u)^{\top}\big(J_C(u)J_C(u)^{\top}\big)^{-1}J_C(u). \tag{72}$$

### C.2  REGULARIZATION AND STABLE IMPLEMENTATION

In practice $A = J_C J_C^{\top}$ can be ill-conditioned/near-singular (e.g., near singular configurations). We use Tikhonov-regularized inversion

$$A_{\epsilon}^{-1} = \big(J_C J_C^{\top} + \epsilon I\big)^{-1}, \qquad \epsilon \in [10^{-8}, 10^{-6}], \tag{73}$$

which bounds the condition number and avoids division by near-zero eigenvalues. For structured constraints:

- **Mass/Momentum conservation:** $\Pi = I - \frac{1}{N}\mathbf{1}\mathbf{1}^{\top}$ ($O(d)$ time via mean removal).
- **Dirichlet boundary:** $\Pi = I - S_b^{\top}S_b$ where $S_b$ selects boundary DOFs (diagonal mask).
- **Divergence-free:** Helmholtz–Hodge projection via Poisson solve (FFT or multigrid).

### C.3 Distributional Continuity on the Manifold

Let $\rho_t$ be the density on $\mathcal{M}$. For the constrained tangent vector field $v_{\mathcal{M}}(u, t) = \Pi(u)\, v_\theta(u, t)$, the continuity equation on $\mathcal{M}$ reads

$$\partial_t \rho_t(u) \;+\; \mathrm{div}_{\mathcal{M}}\big(\rho_t(u)\, v_{\mathcal{M}}(u, t)\big) \;=\; 0. \tag{74}$$

Manifold invariance follows since $J_C(u)\, v_{\mathcal{M}}(u, t) = 0$ and $C(u_0) = 0 \Rightarrow C(u_t) \equiv 0$.

## D  Experimental Setup Details

We evaluate the proposed *Flow-based Automatic Neural Operator with Hard Physical Constraints* on a suite of PDE benchmarks covering advection, conservation laws with shocks, elliptic flow in porous media, and incompressible fluid dynamics. Datasets are split into train/validation/test as $1000/100/100$.

**Model backbone.** We parameterize a time-dependent vector field $g_\theta(u, t)$ as a residual flow network (stacked blocks with pointwise MLPs and spectral/FFT-style convolutions) that transports a simple prior to the solution distribution. To recover high-frequency details possibly removed by hard projections, we add a lightweight *residual compensation branch*: an encoder $E_\phi$ embeds the orthogonal residual into a latent $z$, and a decoder $B_\theta$ with a gain $\alpha_\phi(z)$ injects it back into the tangent dynamics (Sec. C).

We use $L=4$ blocks with width $\approx 64$ for 1D tasks and $L=6$ with channels $\approx 128$ for 2D tasks. Blocks share some weights across scales; multi-resolution spectral modes improve global coupling.

**Training.** We minimize mean squared error (MSE) between model-predicted velocity fields and target fields under *hard* physical projections (no extra penalty terms are needed for constraints). We use Adam/AdamW with initial learning rate $10^{-3}$ and cosine/step decay to $\approx 10^{-4}$, batch size 16–32, and 300–500 epochs. Validation relative-$L^2$ is monitored to select the best checkpoint.

**Compute and cost.** All experiments are performed on a single RTX 5090 GPU . Per iteration, hard projection adds $\sim 15\%$ wall-clock over an operator-only baseline; most of this comes from divergence-free (Poisson) or structured projections. The residual compensation branch adds $\lesssim 1\%$ parameters and negligible time. Inference remains fast (tens of milliseconds per sample), and long rollouts are stable due to strict constraints.

### D.1  Implementation Details Across Benchmarks

#### D.1.1  1D Burgers' Equation

This is a fundamental nonlinear partial differential equation (PDE) in fluid mechanics and mathematical physics, often serving as a standard benchmark for assessing a model's capability to capture dissipative wave phenomena like shock waves and turbulence. This equation incorporates the nonlinear advection term ($u\frac{\partial u}{\partial x}$) and the viscous diffusion term ($\nu\frac{\partial^2 u}{\partial x^2}$), expressed as:

$$\frac{\partial u}{\partial t} + u\frac{\partial u}{\partial x} = \nu\frac{\partial^2 u}{\partial x^2}, \quad x \in (0, 1), t \in (0, 1] \tag{75}$$

where $u(x, t)$ is the velocity field and $\nu$ is the viscosity coefficient.

To generate the data for model training and evaluation, we performed numerical simulations with the following experimental setup:

- **Initial Condition**: We use a smooth sinusoidal function as the initial state: $u(x, 0) = -\sin(\pi x)$.
- **Boundary Conditions**: We impose periodic boundary conditions on the spatial domain, i.e., $u(0, t) = u(1, t)$.
- **Viscosity Coefficient**: Set to $\nu = 0.001$. This low viscosity leads to the formation of discontinuous shock waves in the velocity field, which significantly increases the complexity and challenge of long-term prediction.

- **Spatiotemporal Discretization**: The spatial domain $\Omega = (0,1)$ is discretized into $N_x = 512$ equidistant grid points. The time step is set to $\Delta t = 0.01s$.

Based on the setup above, our learning task is defined as follows. The velocity field $u(x,t)$ at each time step is represented by a discrete state vector $\mathbf{u}_t \in \mathbb{R}^{512}$. The objective is to learn an operator $\mathcal{G}$ that maps the sequence of the first 20 time steps to the sequence of the subsequent 20 time steps. This task is formulated as:

$$\mathcal{G} : \{\mathbf{u}_t\}_{t=1}^{20} \mapsto \{\mathbf{u}_t\}_{t=21}^{40} \tag{76}$$

### D.1.2 1D ADVECTION EQUATION

This is a fundamental linear partial differential equation (PDE) that describes the transport of a scalar quantity. Its ideal analytical solution implies that the initial profile should translate without any change in shape or amplitude (i.e., zero dissipation and dispersion). The equation is expressed as:

$$\frac{\partial u}{\partial t} + c\frac{\partial u}{\partial x} = 0, \quad x \in (0,1), t \in (0,1) \tag{77}$$

where $u(x,t)$ is the transported scalar field and $c$ is the constant wave speed.

Our specific experimental setup is as follows:

- **Initial Condition**: A Gaussian pulse is used as the initial state: $u(x,0) = \exp\left(-\frac{(x-0.5)^2}{2\sigma^2}\right)$, with the pulse width set to $\sigma = 0.1$.

- **Boundary Conditions**: Periodic boundary conditions are imposed on the spatial domain, i.e., $u(0,t) = u(1,t)$.

- **Wave Speed**: Set to a low value of $c = 0.1$. This provides a stringent stress test for the model's ability to suppress accumulated numerical dissipation over long-term integration, thereby preserving physical conservation laws.

- **Spatiotemporal Discretization**: The spatial domain $\Omega = (0,1)$ is discretized into $N_x = 64$ equidistant points. The time step is set to $\Delta t = 0.01s$.

For this task, we define the solution field at each time step as a discrete state vector $\mathbf{u}_t \in \mathbb{R}^{64}$. The objective is to learn an operator $\mathcal{G}$ that maps the solution at the initial time step, $\mathbf{u}_0$, to the solution sequence for the subsequent 20 time steps, $\{\mathbf{u}_t\}_{t=1}^{20}$. This task can be formulated as:

$$\mathcal{G} : \{\mathbf{u}_0\} \mapsto \{\mathbf{u}_t\}_{t=1}^{20} \tag{78}$$

### D.1.3 2D DARCY FLOW

2D Darcy Flow is a more complex steady-state problem. This problem is significant in fields like hydrogeology and petroleum engineering, describing fluid flow through a porous medium. It is governed by an elliptic partial differential equation:

$$-\nabla \cdot (k(\mathbf{x})\nabla u(\mathbf{x})) = f(\mathbf{x}), \quad \mathbf{x} \in (0,1)^2 \tag{79}$$

where $u(\mathbf{x})$ is the pressure or potential field to be solved, $k(\mathbf{x})$ is the given heterogeneous permeability field (the input), and $f(\mathbf{x})$ is a source term.

Unlike the previous time-evolution problems, this is a steady-state task. We approach it using a pseudo-time method, reformulating the problem as a dynamical system that evolves from an initial guess to a steady-state solution. The model learns the right-hand side of this system, which is a differential operator conditioned on the permeability $k$.

The specific experimental configuration is as follows:

- **Equation Setup**: For simplicity, the source term is set to a constant, $f(\mathbf{x}) = 1.0$.

- **Boundary Conditions**: We impose zero Dirichlet boundary conditions on all boundaries of the spatial domain $\Omega = (0,1)^2$, i.e., $u = 0$ on $\partial\Omega$.

- **Data Discretization**: The spatial domain is discretized onto a uniform $128 \times 128$ grid.

- **Dataset**: We use a dataset consisting of 1000 training and 100 testing samples. Each sample contains a permeability field $k(\mathbf{x})$ and its corresponding numerical solution $u(\mathbf{x})$.

Our learning task is to learn an operator $\mathcal{G}$ that maps any given permeability field $k$ to the corresponding steady-state pressure field $u$:

$$\mathcal{G} : \mathbf{k} \mapsto \mathbf{u} \tag{80}$$

where $\mathbf{k}, \mathbf{u} \in \mathbb{R}^{41 \times 41}$ are the discretized permeability and pressure fields, respectively.

### D.1.4  2D ADVECTION EQUATION

To study the model's ability to handle advection-dominated partial differential equations, we consider a linear advection equation with a variable coefficient. While this equation is 1D in space, we represent its solution as a 2D spatiotemporal field (space × time) for neural operator input. Its mathematical form is as follows:

$$\frac{\partial s}{\partial t} + u(x) \frac{\partial s}{\partial x} = 0, \quad (x, t) \in (0, 1) \times (0, 1) \tag{81}$$

where $s(x, t)$ is the scalar field to be solved, and $u(x)$ is a non-constant advection speed that depends only on the spatial position $x$ (i.e., the variable coefficient).

The detailed configuration for this experiment is described as follows:

- **Initial and Boundary Conditions**: The equation is defined with specific initial and boundary conditions:
  - Initial Condition: $s(x, 0) = f(x) = \sin(\pi x)$
  - Boundary Condition: $s(0, t) = g(t) = \sin(\frac{\pi}{2} t)$
- **Data Discretization**: The original dataset had a spatial resolution of 1024 and a temporal resolution of 201. To accommodate model training, we downsampled along both the time and space dimensions. The final grid resolution used for the experiment is: 40 points in the spatial dimension ($N_x = 40$) and 41 points in the temporal dimension ($N_t = 41$).
- **Dataset**: The dataset we use contains 1000 training samples and 200 test samples. Each sample represents a complete spatiotemporal evolution.

The learning objective of this experiment is to learn an operator $\mathcal{G}$ that can map from the variable coefficient to its corresponding solution.

- **Input**: The initial condition vector $\mathbf{u}_0 \in \mathbb{R}^{N_x}$ is replicated along the temporal dimension to construct an input tensor $\mathcal{U} \in \mathbb{R}^{N_x \times N_t}$ matching the target output dimensions.
- **Output**: The complete 2D spatiotemporal solution $s(x, t)$ of the physical system under the given initial and boundary conditions.

This learning task can be formally expressed as:

$$\mathcal{G} : u(x, t = 0) \mapsto s(x, t)|_{(0,1) \times (0,1)} \tag{82}$$

On the discretized grid, this mapping is $\mathcal{G} : \mathbf{u} \mapsto \mathbf{S}$, where the input is $\mathbf{u} \in \mathbb{R}^{40 \times 41}$ and the output is $\mathbf{S} \in \mathbb{R}^{40 \times 41}$.

### D.1.5  2D NAVIER-STOKES EQUATION

We conduct experiments on the 2D incompressible Navier-Stokes (NS) equation, a foundational equation in fluid dynamics. We study its vorticity transport form, which is crucial for analyzing complex phenomena such as turbulence.

$$\frac{\partial \omega}{\partial t} + \mathbf{u} \cdot \nabla \omega = \nu \nabla^2 \omega, \quad (\mathbf{x}, t) \in (0, 1)^2 \times (0, T] \tag{83}$$

where $\omega(\mathbf{x}, t)$ is the vorticity field of the fluid, $\mathbf{u}$ is the velocity field, and $\nu$ is the kinematic viscosity of the fluid.

The experimental parameter configuration is as follows:

- **Equation Setup**: We solve the vorticity transport equation in a 2D space with the viscosity coefficient set to $\nu = 1 \times 10^{-5}$. This corresponds to a high-Reynolds-number turbulent state, making long-term prediction extremely challenging.

- **Boundary Conditions**: The fluid domain is $(0, 1)^2$, with periodic boundary conditions applied in both spatial directions.

- **Data Discretization**: The spatial domain is discretized into a uniform $64 \times 64$ grid. In the temporal dimension, we use data from a total of 20 time steps for training and testing.

- **Dataset**: The dataset consists of 1000 training samples and 200 test samples.

The learning task for this experiment is to learn an operator $\mathcal{G}$ capable of predicting the future evolution of the fluid field based on its state over a past period. This is a sequence-to-sequence learning problem.

- **Input**: The complete vorticity field history sequence for the first 10 time steps ($T_{in} = 10$).
- **Output**: The vorticity field evolution sequence for the subsequent 10 time steps ($T_{out} = 10$).

This learning task can be formally expressed as:

$$\mathcal{G} : \{\omega(\mathbf{x}, t)\}_{t=1}^{10} \mapsto \{\omega(\mathbf{x}, t)\}_{t=11}^{20} \tag{84}$$

where the vorticity field $\omega$ at each time step is a 2D field discretized on a $64 \times 64$ grid.

# E   VISUALIZATION PER PDE BENCHMARK

Below we provide *placeholders* indicating where to insert figures. Please replace each box with your result images. No captions are provided by request; the text in the box is only a placement note.

## E.1   1D LINEAR ADVECTION

The solution translates without distortion and preserves total mass.

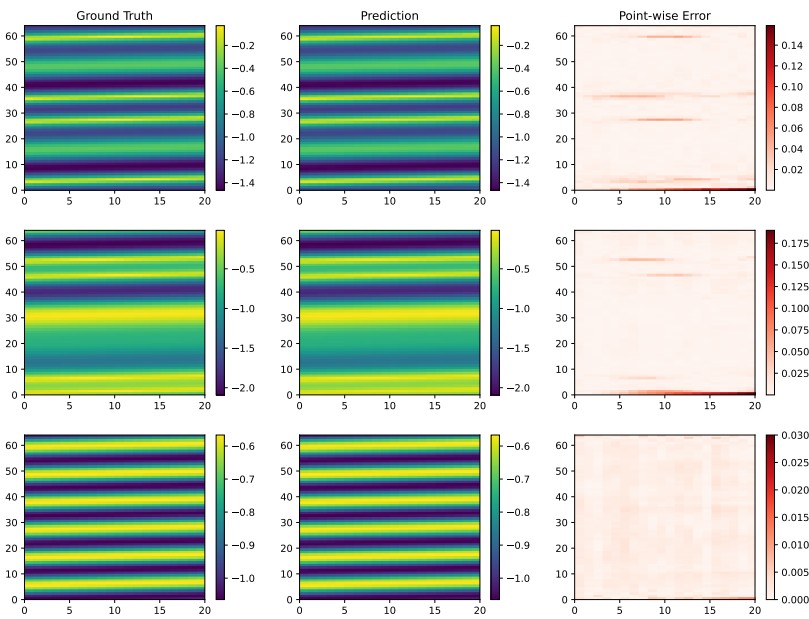

Figure 7: Spacetime predictions for 1D Advection: Each column shows (left) ground truth $u(x, t)$, (center) model prediction $\hat{u}(x, t)$, and (right) absolute error $|u - \hat{u}|$ across the full spacetime domain ($x \in [0, L]$, $t \in [0, 20]$s)(output domain $t = 0$ corresponds to $t = 20$s in input coordinates). Rows display results for distinct initial conditions.

## E.2 1D BURGERS WITH SHOCKS

Residual compensation is crucial to preserve shock steepness while enforcing conservation.

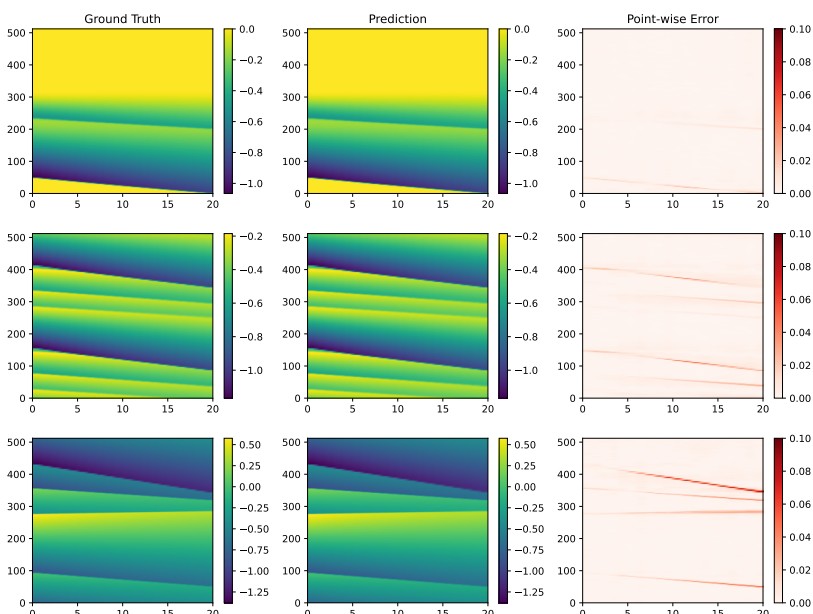

Figure 8: Spacetime predictions for Burgers' advection: Each column shows (left) ground truth $u(x, t)$, (center) model prediction $\hat{u}(x, t)$, and (right) absolute error $|u - \hat{u}|$ across the full spacetime domain ($x \in [0, L]$, $t \in [0, 20]$s). Rows display results for distinct initial conditions.

## E.3 2D DARCY FLOW

Steady elliptic problem with Dirichlet boundary; predictions match pressure contours closely.

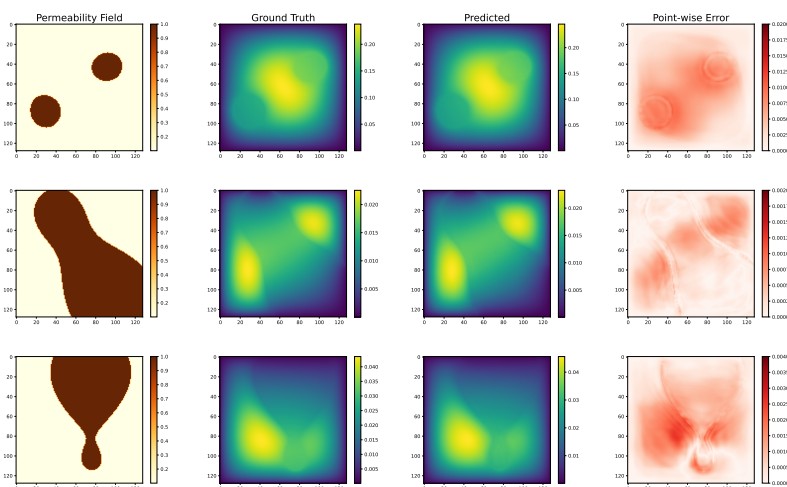

Figure 9: 2D Darcy flow predictions: Columns show (left) input permeability field $k(x, y)$, (center left) reference pressure $p_{\text{ref}}(x, y)$, (center right) predicted pressure $\hat{p}(x, y)$, and (right) absolute error $|p_{\text{ref}} - \hat{p}|$. Rows display three distinct test cases with varying input permeability distributions (top to bottom).

### E.4 2D INCOMPRESSIBLE NAVIER–STOKES

We test long rollouts and incompressibility.

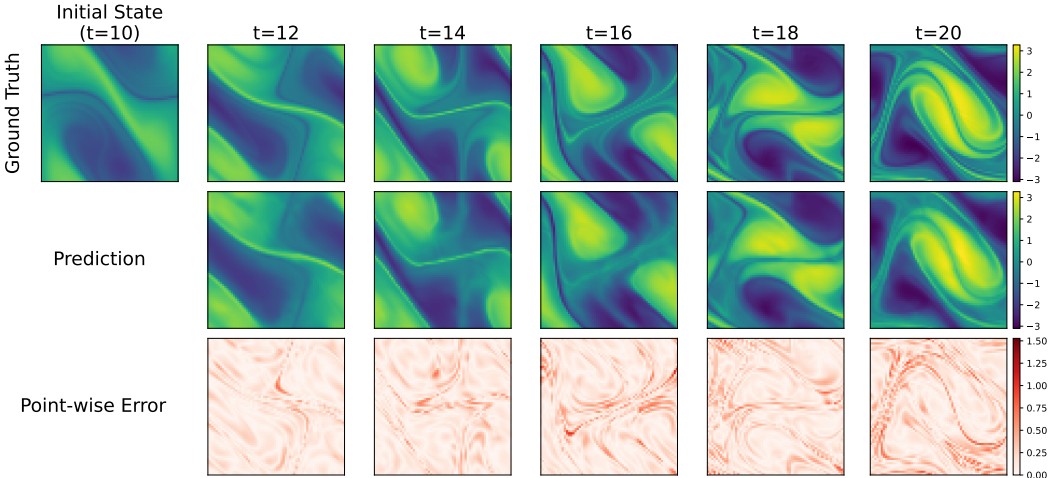

Figure 10: Vorticity dynamics benchmark: Temporal evolution predicted from $t_0 = 0$ to $t_{in} = 10$ initial condition, showing vorticity field $\omega(x, y, t)$ at $t = \{13, 16, 19\}$ (top to bottom). Columns show (left) reference solution $\omega_{\text{ref}}$, (middle) predicted field $\hat{\omega}$, and (right) absolute error $|\omega_{\text{ref}} - \hat{\omega}|$.

## F ABLATION RESULTS

Table 2: Quantitative comparison of the full PMFM model and its ablated version without physical constraints. We report the final Mean Squared Error (MSE) and the average violation of a key conservation law (e.g., Energy Drift). Lower values are better for both metrics.

| Model Variant | Final MSE |
|---|---|
| PMFM (Full Model) | 4.68e-05 |
| w/o Physical constraints | 2.73e-04 |

## G ADAPTIVE CONSTRAINT PROJECTION: FULL PSEUDOCODE

In this section, we provide the complete pseudocode of the proposed Adaptive Constraint Projection Framework. The framework integrates flow-matching training with adaptive constraint enforcement and a compensation pathway, ensuring that the learned dynamics remain consistent with the physical manifold. We present three algorithms: (i) the training loop for Physics-Manifold Flow Matching (PMFM), (ii) an alternative tabular description of constraint-gated updates, and (iii) the inference/prediction routine using the trained model.

**Algorithm A.1** *Adaptive Constraint Projection Framework (training-time flow-matching).*

---

**Algorithm 1** Training the Physics-Manifold Flow Matching (PMFM) Model

---

**Require:**

> Neural network parameters $\Theta = \{\theta, \phi\}$ for networks $g_\theta, B_\theta, \alpha_\phi, E_\phi$.
> Data distribution $p_{\text{data}}(\mathbf{u})$ and prior distribution $p_0(\mathbf{u})$.
> Physical constraint functions $C(\mathbf{u}) = \{C_i(\mathbf{u})\}_{i=1}^m$.

**Ensure:**

> Optimized network parameters $\Theta^*$.

1: **for** each training step **do**
2:      Sample a mini-batch of data pairs $\{(\mathbf{u}_1^{(i)}, \mathbf{u}_0^{(i)})\}_{i=1}^B \sim p_{\text{data}} \times p_0$.
3:      Sample a time-step $t \sim U(0, 1)$.
4:      **for** each pair $(\mathbf{u}_1, \mathbf{u}_0)$ in the mini-batch **do**
5:          Construct the straight path: $\mathbf{u}_t \leftarrow (1 - t)\mathbf{u}_0 + t\mathbf{u}_1$.
6:          Define the target vector field: $\mathbf{v}^* \leftarrow \mathbf{u}_1 - \mathbf{u}_0$.
7:          Compute the constraint Jacobian $\mathbf{J}_C(\mathbf{u}_t)$.
8:          Form the projection operator $\Pi_{T_{\mathbf{u}_t}\mathcal{M}}$.
9:          Compute residual: $\mathbf{r}_t \leftarrow (\mathbf{I} - \Pi_{T_{\mathbf{u}_t}\mathcal{M}})\mathbf{v}^*$.
10:        Encode residual into latent code: $\mathbf{z}_t \leftarrow E_\phi(\mathbf{r}_t)$.
11:        Predict compensated field: $\mathbf{v}_{\text{comp}} \leftarrow g_\theta(\mathbf{u}_t, t) + B_\theta(\mathbf{u}_t, t)\alpha_\phi(\mathbf{z}_t)$.
12:        Compute loss: $\mathcal{L}_{\text{sample}} \leftarrow \|\mathbf{v}_{\text{comp}} - \mathbf{v}^*\|^2$.
13:      **end for**
14:      Compute batch loss: $\mathcal{L}(\Theta) \leftarrow \frac{1}{B}\sum_{i=1}^B \mathcal{L}_{\text{sample}}^{(i)}$.
15:      Update $\Theta \leftarrow \Theta - \eta\nabla_\Theta \mathcal{L}(\Theta)$.
16: **end for**
17: **return** $\Theta^*$.

---

**Algorithm A.2** *Tabular description of adaptive gating and projection.*

---

**Inputs:** constraint library $\{C_k(u) = 0\}_{k=1}^M$ with Jacobians $\{J_{C_k}\}$; projection builders; prior $p(u_0)$; reference trajectories $\{u^\star(t)\}$
**Outputs:** trained parameters $\Theta = \{\theta, \phi\}$

---

1:   initialize $\Theta$
2: **for** iter $= 1, \ldots$ **do**
3:    sample reference trajectory $\{u^\star(t)\}_{t=0}^T$ and $u(0) \sim p$
4:    **for** $t = 0, \ldots, T - 1$ **do**
5:      $S \leftarrow \text{GatingNetwork}(u(t), t)$                                // active constraints
6:      $v_{\text{base}} \leftarrow g_\theta(u(t), t)$
7:      **if** $S \neq \varnothing$ **then** build $J_S$ and $\Pi_S$; $v_{\text{proj}} \leftarrow \Pi_S v_{\text{base}}$
8:      residual $r \leftarrow v_{\text{base}} - v_{\text{proj}}$; latent $z \leftarrow E_\phi(r)$
9:      $v_{\text{comp}} \leftarrow v_{\text{proj}} + B_\theta(u(t), t)\,\alpha_\phi(z)$
10:     $u(t+1) \leftarrow u(t) + (\Pi_S v_{\text{comp}}$ if $S \neq \varnothing$ else $v_{\text{comp}})$
11:    compute loss $L(\Theta)$; update $\Theta$
12: **end for**

---

**Algorithm A.3** *Inference/Prediction with a trained PMFM model.*

---

**Algorithm 2** Inference/Prediction with a Trained PMFM Model

---

**Require:** Trained parameters $\Theta^* = \{\theta^*, \phi^*\}$, prior $p_0(u)$, constraints $C(u)$, number of ODE steps $N$

**Ensure:** Final state $u_{\text{out}}$ with $C(u_{\text{out}}) = 0$

1: Sample initial state: $u_{\text{current}} \sim p_0(u)$
2: Set $\Delta t \leftarrow 1/N$
3: **for** $k = 0, \ldots, N - 1$ **do**
4:      $t \leftarrow k \cdot \Delta t, z \leftarrow \mathbf{0}$
5:      $v_{\text{comp}} \leftarrow g_{\theta^*}(u_{\text{current}}, t) + B_{\theta^*}(u_{\text{current}}, t)\alpha_{\phi^*}(z)$
6:      Compute $J_C(u_{\text{current}})$ and $\Pi_{T_{u_{\text{current}}}\mathcal{M}}$
7:      $v_{\mathcal{M}} \leftarrow \Pi_{T_{u_{\text{current}}}\mathcal{M}}(v_{\text{comp}})$
8:      Update: $u_{\text{current}} \leftarrow u_{\text{current}} + v_{\mathcal{M}} \cdot \Delta t$
9: **end for**
10: $u_{\text{out}} \leftarrow \Pi_{\mathcal{M}}(u_{\text{current}})$
11: **return** $u_{\text{out}}$

---

## H  NOTES ON PRACTICAL IMPLEMENTATION

**Autodiff with projections.** All projection operators are differentiable w.r.t. $u$ (and task parameters), so we keep them inside the computational graph. For structured operators (masks, mean-removal) gradients are trivial; for Poisson solves we treat the linear solve as a differentiable layer.

**Constraint conflicts.** If multiple constraints are redundant/incompatible, $J_C J_C^\top$ may be ill-conditioned. We avoid conflicts via gating (§G) and small $\epsilon$-regularization; if necessary, we prioritize primary constraints and add soft penalties for secondary ones.

**Complexity.** Mass/momentum/Dirichlet projections are $O(d)$. Divergence-free is dominated by one Poisson solve per step (near-linear with FFT/multigrid). Overall overhead was modest ($\sim 15\%$).

## I  LLM USAGE

A Large Language Model (LLM) was utilized to assist with the linguistic polishing of this manuscript. The model's contributions were confined to improving language clarity, grammar, and overall readability. All scientific concepts, methodologies, and analyses were developed exclusively by the authors.

# J ADDITIONAL EXPERIMENTS AND ANALYSIS

In this section, we provide detailed visualizations and experimental results to support our rebuttal.

## I.1 A. TRAINING-INFERENCE MISMATCH ANALYSIS

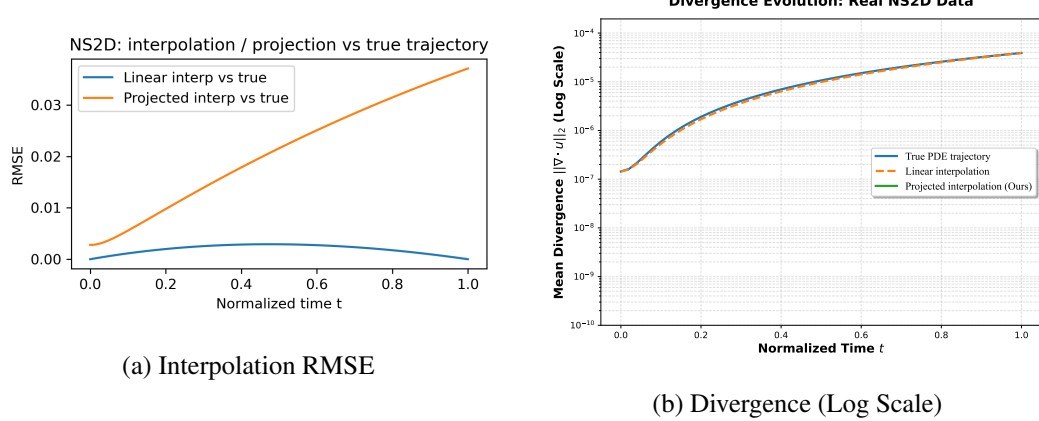

**Burgers1D: Linear interpolation vs true trajectory**

Figure R1: The RMSE curve for Burgers 1D shows a significant deviation (RMSE $\approx 1.0$) between the linear interpolation path used during training and the true physical trajectory, confirming the distribution mismatch.

## I.2 B. DIVERGENCE ANALYSIS ON NS2D

Figure R2: (a) RMSE comparison showing that projected interpolation deviates from the true numerical solution. (b) Divergence curves showing that our projected method maintains physical consistency at machine precision ($\sim 10^{-9}$), while both the true numerical solution and linear interpolation drift significantly.

(a) Interpolation RMSE

(b) Divergence (Log Scale)

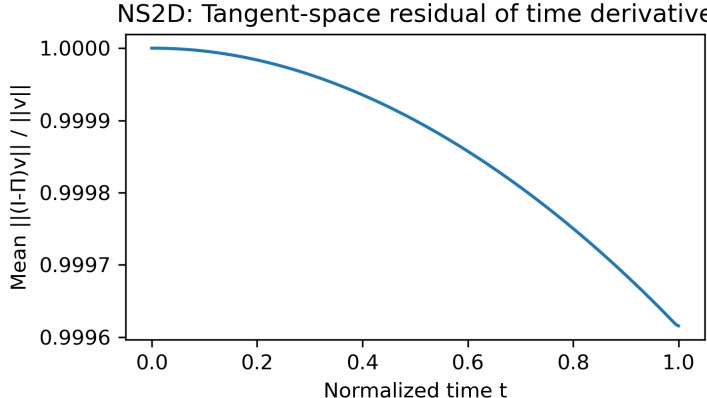

Figure R3: The ratio approaches 1.0, indicating that the constraint-enforcing forces (pressure gradients) dominate the vector field magnitude, necessitating the projection operator to extract the valid tangential dynamics.

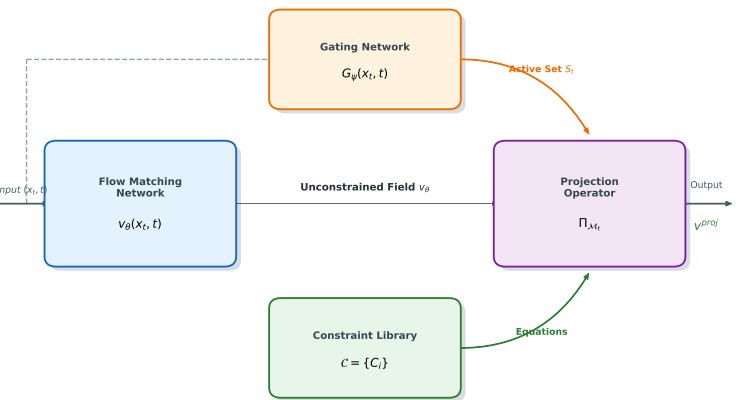

Figure R4: Illustration of the PMFM framework where the Gating Network dynamically selects constraints to construct the manifold $\mathcal{M}_t$, onto which the vector field $v_\theta$ is projected.

We validated the scalability of PMFM on the 3D Diffusion dataset ($32^3$) and compared it with the ECI Sampling baseline.

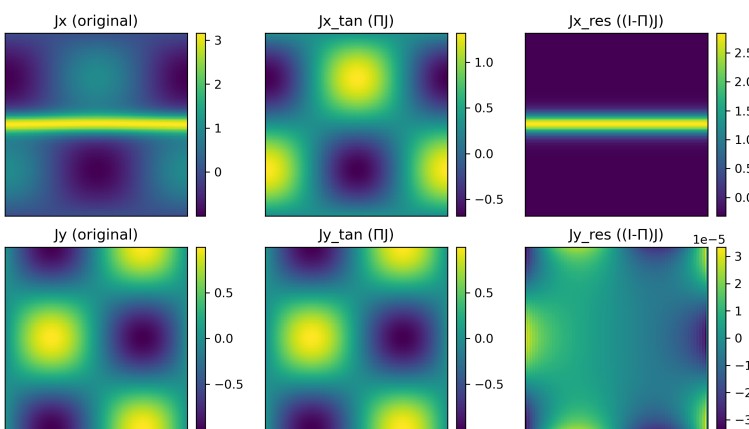

Figure R5: Visualizations of the original vector field $J$, its tangential component $\Pi J$, and the residual $(I - \Pi)J$. Note that the residual contains fine-grained structural information.

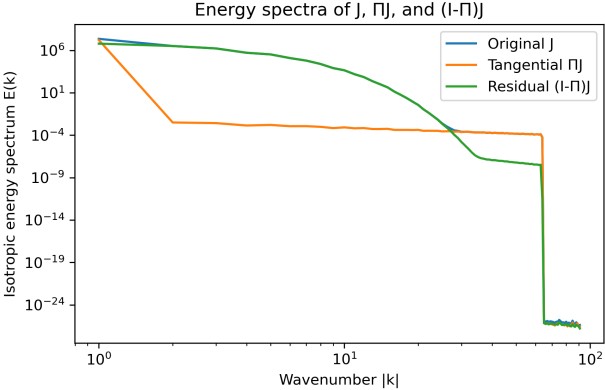

Figure R6: The energy spectrum confirms that the residual component $(I - \Pi)J$ (Green) dominates the high-frequency domain, validating the need for GGM to recover these details.

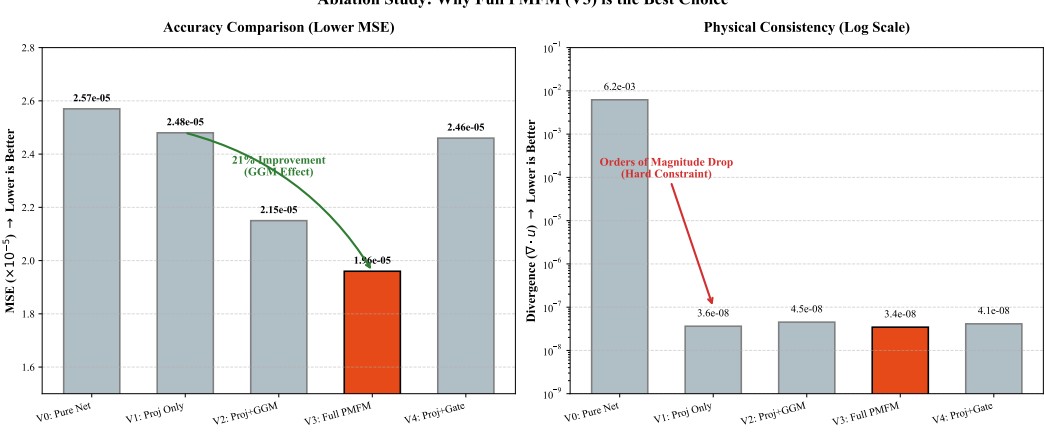

Figure R7: Side-by-side comparison showing that Full PMFM (V3) achieves the best accuracy (MSE) while maintaining machine-precision physical consistency (Divergence).

Table R1: Isolating the contribution of Gating, GGM, and Projection. **V3 (Ours)** achieves the best balance of accuracy and physical consistency.

| Variant | MSE ($\phi$) | Div ($\nabla \cdot u$) | Observation |
|---|---|---|---|
| V0: Pure Network | $2.57 \times 10^{-5}$ | $6.22 \times 10^{-3}$ | Low error, but physically inconsistent |
| V1: Proj Only | $2.48 \times 10^{-5}$ | $3.63 \times 10^{-8}$ | Perfect physics, but loses details |
| V2: Proj + GGM | $2.15 \times 10^{-5}$ | $4.50 \times 10^{-8}$ | GGM improves MSE; static constraint OK |
| **V3: Full PMFM** | $\mathbf{1.96 \times 10^{-5}}$ | $\mathbf{3.55 \times 10^{-8}}$ | **Best Accuracy & Physics** |
| V4: Proj + Gate | $2.46 \times 10^{-5}$ | $4.12 \times 10^{-8}$ | Gate correctly identifies active constraints |

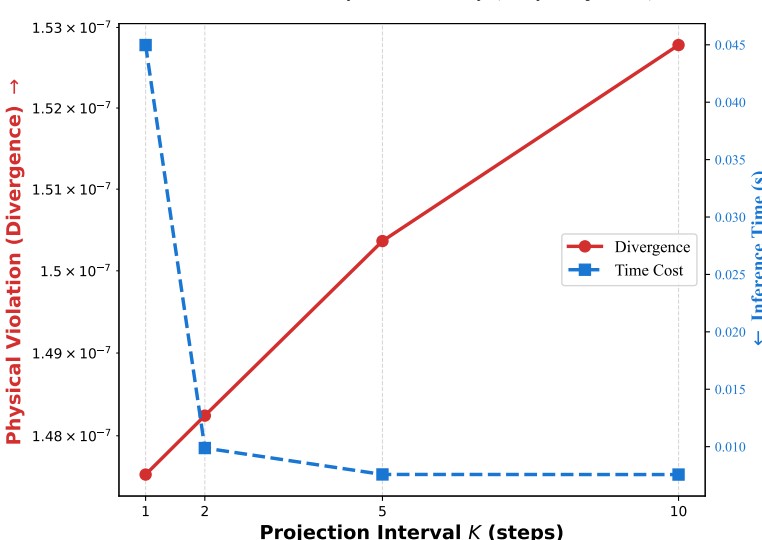

Figure R8: **Accuracy-Efficiency Trade-off**: Effect of projection interval $K$ on inference time and physical violation. Increasing $K$ yields significant speedups (up to 4.5$\times$) with a controlled increase in divergence.

Table R2: Comparing our method vs. training with per-step projection. Our strategy is **3.02$\times$ faster** and achieves lower training loss.

| Metric | Standard (Ours) | Projected Training | Impact |
|---|---|---|---|
| **Time / Epoch** (s) | **0.0939** | 0.2836 | **3.02$\times$ Slower** |
| **Final Train Loss** | $\mathbf{2.0 \times 10^{-6}}$ | $5.0 \times 10^{-6}$ | **+150% (Worse)** |

Table R3: Comparison between ECI Sampling and our PMFM. Our method achieves superior accuracy and faster inference.

| Method | MSE ($\downarrow$) | Inference Time (s/step) $\downarrow$ | Improvement |
|---|---|---|---|
| ECI Sampling | $1.07 \times 10^{-3}$ | 0.060 | – |
| **PMFM (Ours)** | $\mathbf{8.92 \times 10^{-4}}$ | **0.056** | **+17% (Accuracy)** |

