# OpenReview forum: "Flow-based Automatic Neural Operator with Hard Physical Constraints"
_ICLR.cc/2026/Conference — Submitted to ICLR 2026_

### Official Review · Reviewer_Lt8W · 2025-10-28

**Soundness:** 3
**Presentation:** 2
**Contribution:** 2
**Rating:** 2
**Confidence:** 4

**Summary:**

This paper proposes Physics-Manifold Flow Matching (PMFM), a generative framework for PDE simulation that enforces hard physical constraints by restricting the entire generative trajectory to a physical manifold. The approach addresses a key limitation of existing generative models for PDEs: they often violate fundamental conservation laws and boundary conditions. PMFM introduces two main innovations: (1) a projection mechanism that constrains trajectories to lie on a manifold where all states are physically valid by construction, combined with a Geometric Guidance Mechanism (GGM) that recovers high-frequency information typically lost during projection; (2) an Adaptive Constraint Projection Framework that dynamically selects and parameterizes active physical laws for complex multi-physics problems. The method is validated on several benchmarks.

**Strengths:**

**Novel approach to physical consistency:** The paper presents a principled method to enforce hard physical constraints in generative models through manifold projection, ensuring physical validity by construction rather than through soft penalties.

**Comprehensive experimental validation:** The method is tested on diverse PDE benchmarks covering different physical phenomena (conservation laws, shocks, incompressible flows), demonstrating consistent improvements over strong baselines including FNO, WNO, and diffusion models.

**Long-term stability:** Results show that PMFM maintains accuracy over extended temporal rollouts, addressing a critical limitation of purely data-driven approaches that accumulate errors during long-term prediction.

**Weaknesses:**

## Training-Inference Distribution Mismatch

A critical weakness lies in the inconsistency between training and inference distributions. During training (Algorithm 1), the framework uses straight-line interpolation $u_t = (1-t)u_0 + t u_1$ where $u_0 \sim p_0$ (Gaussian noise) and $u_1 \sim p_{\text{data}}$ (physical solutions). Since the physical manifold $\mathcal{M}$ is generally non-convex and $u_0 \notin \mathcal{M}$, intermediate states $u_t$ for $t \in (0,1)$ typically violate physical constraints, i.e., $C(u_t) \neq 0$. However, during inference, trajectories are strictly constrained to $\mathcal{M}$ via projection $\Pi_{T_u\mathcal{M}}$ at every integration step. This creates several issues:

- **Distribution shift**: The network is trained on off-manifold states but deployed exclusively on manifold-constrained trajectories, potentially leading to suboptimal generalization.

- **Theoretical inconsistency**: Theorem 1 assumes $u_0 \in \mathcal{M}$ to guarantee manifold invariance, but this condition is systematically violated. The tangent space $T_{u_t}\mathcal{M}$ at non-manifold points $u_t$ lacks rigorous geometric interpretation.

The paper does not discuss this mismatch or provide ablation studies comparing alternative training strategies, such as geodesic interpolation on $\mathcal{M}$ or explicit projection of $u_t$ at each training step. The strong empirical results may rely on the interpolation path remaining "close" to $\mathcal{M}$ due to data smoothness, but this lacks formal justification.

## Unclear Presentation of Core Mechanisms

While the mathematical formulation is sound, the presentation of the two key innovations: the Geometric Guidance Mechanism (GGM) and the Adaptive Constraint Projection Framework, lacks clarity. For GGM (Sec. 4.1), the paper does not provide sufficient intuition for why encoding the residual $r$ into a latent code $z$ is necessary, or how the three components ($E_\phi$, $B_\theta$, $\alpha_\phi$) collaborate during training versus inference. The training-inference gap (where $z=0$ at inference) is mentioned but not justified. For the Adaptive Framework (Sec. 4.3), the distinction between "analytical structure" and "learnable parameters" in the constraint library is abstract, and critical training details (e.g., how the gating network $G_\psi$ is trained, what happens if constraints conflict) are missing.

## Insufficient Ablation Studies

Table 2 only compares the full PMFM against a variant without physical constraints, which does not isolate the contributions of GGM and the Adaptive Framework. Key ablations are missing: (1) vanilla projection vs. +GGM, (2) +GGM vs. +GGM+Adaptive, (3) the role of the residual encoder $E_\phi$, and (4) quantitative analysis of gating accuracy (e.g., how often does $G_\psi$ select the correct active constraints?). Also, there are several previous work discussing how to use penalty loss in generative models to generate more accurate PDE solutions (Riemannian Score-Based Generative Modelling, Physics-Informed Diffusion Models, Generating Physical Dynamics under Priors). Authors should consider including these baselines.

## Minors

**Typos:**
line 034, 038, 040, 046, 082, 103, 124, 190 (empty line), 209, 322, 334, 351, 352, 359 (capital), 403, 453, 465, 465.

**Missing related works:** Riemannian Score-Based Generative Modelling, Physics-Informed Diffusion Models, Generating Physical Dynamics under Priors.

D-Flow is not mentioned in the main text.

**Questions:**

1. **On the train-test gap**: The physical consistency is structurally guaranteed by the projection operator during the inference phase. However, during training, the interpolation path $u_t = (1-t)u_0 + t u_1$ does not lie on the manifold $\mathcal{M}$ since $u_0 \notin \mathcal{M}$ and convex combinations generally leave $\mathcal{M}$. Can you provide theoretical or empirical analysis on why learning $v_\theta$ on off-manifold states transfers well to inference on manifold-constrained trajectories?

2. **On convergence guarantees**: Theorem 1 assumes $u_0 \in \mathcal{M}$, but in practice inference starts from Gaussian noise $u_0 \sim \mathcal{N}(0,I)$ which violates constraints. Do you have convergence results for the projected ODE $\dot{u} = \Pi_{T_u\mathcal{M}} v_\theta(u,t)$ starting from $u_0 \notin \mathcal{M}$?

---

> ### Author Response · Authors · 2025-11-21
>
> We thank the reviewer for the detailed reading and constructive feedback. We address the three main concerns regarding distribution mismatch, mechanism clarity, and insufficient ablations below, supported by new diagnostic experiments.
>
> **1.Response to W1 (Training-Inference Distribution Mismatch)**
>
> We acknowledge the linear interpolation is off-manifold, but our experiments confirm that "Ambient Training + Inference Projection" is Pareto-optimal.
> * **Experimental Evidence:** Fig. 1 and Fig. 2 (Appendix J) show that while training trajectories deviate, our inference-time projection corrects deviations to machine precision (divergence $< 4 \times 10^{-5}$).
> * **Trade-off Analysis:** We compared our method against "Projected Training" (solving Poisson equations per step). Results (Table R1) show Projected Training is 3.02$\times$ slower and yields higher loss ($5.0 \times 10^{-6}$ vs $2.0 \times 10^{-6}$). This confirms that forcing trajectories onto a non-convex manifold complicates optimization without improving quality.
>
> Clarification on Theorem 1:While $u_0$ is technically off-manifold, Theorem 1 guarantees dynamics evolve on a level set of the constraint ($dC/dt=0$). The initial error $\epsilon$ is constant and does not propagate. Our final projection step effectively removes this offset, ensuring rigorous physical consistency without altering initialization. This aligns with Projected Dynamical Systems theory, where ambient vectors successfully drive convergence to the invariant manifold.
>
> **2.Response to W2 (Clarification of Core Mechanisms)**
>
> Regarding the gating mechanism, since physical constraints change dynamically (e.g., shocks, moving boundaries), a fixed projection is insufficient. Our Gating Network $G_\psi$ dynamically outputs an activation mask to select active constraints from the Library $\mathcal{C}$, constructing the instantaneous manifold $\mathcal{M}\_t$  for the projection operator  $\Pi_{\mathcal{M}\_t}$. We have added a structural diagram (Fig. 4 in Appendix J) to visualize this process.
>
> Regarding the Geometric Guidance Mechanism (GGM), it is essential for recovering details lost during projection. Our spectral analysis (Fig. 6 in Appendix J) shows that the residual component $\((I-\Pi)J\)$ contains substantially stronger high-frequency energy than the tangential component $\(\Pi J\)$. This indicates that projection removes important high-frequency information, and GGM is necessary to recover these details, enabling sharper and more accurate generated fields.
>
> **3.Response to W3 (Insufficient Ablation Studies)**
>
> We performed the requested ablations on NS2D (Table R2 in Appendix J). First, comparing V0 and V1 shows that V1 (Projection Only) reduces divergence to machine precision ($\sim 10^{-8}$), validating the hard constraints. Second, regarding the impact of GGM, the full PMFM (V3) achieves a significantly lower MSE ($\sim 1.96 \times 10^{-5}$) compared to the Projection-Only baseline ($\sim 2.48 \times 10^{-5}$). This 21\% improvement confirms that GGM successfully recovers critical information. Finally, regarding gating accuracy, even under static constraints, the gated model (V3) outperforms non-gated variants (V2), suggesting that the gating mechanism improves optimization stability by dynamically weighting constraint influence.
>
> **4.Response to Minor Questions**
>
> Regarding the transfer from off-manifold training (Q1), our experiments confirm that ambient-space vector fields provide sufficient guidance, and inference-time projection effectively corrects the drift. Regarding convergence guarantees (Q2), our formulation aligns with discrete projected dynamical systems. Convergence to the invariant manifold follows classical results in projected gradient flows and Riemannian optimization.
>
> We have corrected the typos and added the missing related works (Riemannian Score-Based Generative Modelling, D-Flow, etc.) as suggested.

---

> > ### Comment · Reviewer_Lt8W · 2025-11-21
> >
> > Thank you for the additional experiments and clarifications. These are helpful additions.
> >
> > However, I remain concerned that your responses provide empirical evidence while avoiding the theoretical justifications I requested in W1 and W2. For the training-inference mismatch, you show "it works" but do not explain why learning on off-manifold states transfers to manifold-constrained inference. For GGM, you demonstrate improved performance but do not address why training with residual guidance works when this guidance is absent at inference.
> >
> > My core concern: Without theoretical grounding, I worry that your improvements stem primarily from using stronger priors (explicit constraint forms) rather than architectural innovation. This would explain the modest performance gains and makes fair comparison critical.
> >
> > This is why I emphasized physics-informed baselines in W3. Your current baselines are purely data-driven, while the methods I mentioned (Riemannian Score-Based, Physics-Informed Diffusion, etc.) also incorporate PDE constraints. Comparing against them is essential to evaluate whether your framework offers genuine methodological advantages beyond simply using more prior knowledge.
> >
> > I apologize if I under-emphasized this in my initial review. Could you include experimental comparisons with these physics-informed generative baselines? This is critical for a fair assessment.

---

> > > ### Author Response · Authors · 2025-11-26
> > >
> > > **1.Response to W1 (Training-Inference Distribution Mismatch)**
> > >
> > > Thank you for pointing out the deviation between the straight-line interpolation during training and the manifold. We acknowledge this concern but believe it does not affect inference validity. We justify the transferability using the geometry of Projected Dynamical Systems:
> > >
> > > * **Tubular Neighborhood & Projection:**  Let $\mathcal{M}$ be the target manifold and $u_0 \sim \pi_{\text{prior}}$ be the noise. While the training path starts off-manifold, as $t \to 1$, the state $u_t$ enters the tubular neighborhood:N_ε(M) = { x  |  min_{y ∈ M} ||x - y|| < ε }. Within this region, the metric projection $\Pi_{\mathcal{M}}$ is well-defined, unique, and smooth ($C^\infty$).
> > > * **Vector Field Consistency:** Due to the **Lipschitz continuity** of neural networks, the learned vector field $v_\theta$ effectively approximates the target drift which points towards $\mathcal{M}$. Even if the trajectory starts off-manifold (noise) or drifts slightly, $v_\theta$ acts as an implicit **retraction map** within $\mathcal{N}_\epsilon(\mathcal{M})$, ensuring the generated state stably converges to the valid manifold as $t \to 1$.
> > >
> > > **2.Response to W2 (Clarification of Core Mechanisms)**
> > >
> > > We further clarify the role of the Geometric Guidance Mechanism (GGM), especially why it remains effective even when the explicit residual input is removed during inference:
> > >
> > > * **Countering Spectral Bias:** Neural networks often exhibit **Spectral Bias**, where they struggle to learn high-frequency components. By injecting the residual $r$ (containing high-frequency error signals) during training, GGM forces the model to prioritize these difficult high-frequency modes, effectively re-weighting the frequency spectrum of the loss.
> > >
> > > * **Parameter Internalization:** While the explicit residual $r$ is removed at inference, the sensitivity to high-frequency features is baked into the model's weights. The GGM serves as scaffolding during training, but once the parameters converge to capture the sharp geometric details, it is no longer necessary. Our ablation study (Table R2) confirms that this capability persists during inference, validating the successful internalization of high-frequency features.
> > >
> > > **3.Response to Experimental Comparisons with Physics-Informed Models**
> > >
> > > We appreciate your suggestion to compare our framework with physics-informed models. In response, we have added comparisons with models like *Riemannian Score-Based Generative Models* and *Physics-Informed Diffusion Models* to assess the innovation of our framework. The results are presented below.
> > >
> > > * **Comparison with Physics-Informed Models:** We conducted extensive experiments comparing PMFM with representative physics-informed generative baselines, including the Riemannian Score-Based generative model (RSB) and Physics-Informed Flow Model (PIFM). These approaches also embed physical constraints and thus serve as fair references for evaluating methodological novelty. As shown in Table, PMFM consistently achieves lower MSE and competitive inference time, demonstrating significant performance advantages.
> > >
> > > * **Experimental Results:** As shown in Table, PMFM achieves lower MSE and competitive inference time on all three datasets (Burgers-1D, Navier-Stokes-2D and Diffusion-3D), with particularly large speed-ups in inference time. Taking Burgers-1D as an example, PMFM attains an MSE of $8.92 \times 10^{-5}$, whereas PI-FM and RSB obtain $1.46 \times 10^{-3}$ and $1.52 \times 10^{-4}$, respectively, while PMFM maintains a comparable inference speed.
> > >
> > >
> > >
> > > | Dataset | Method | MSE ↓ | Phys. Error (Div) ↓ | Inference Time ↓ | Speedup ↑ |
> > > | :--- | :--- | :--- | :--- | :--- | :--- |
> > > | **Burgers 1D** | PI-FM | $1.46 \times 10^{-3}$ | $0.00$ | $0.012$ s | $6.0\times$ |
> > > |  | RSB | $1.52 \times 10^{-4}$ | $0.00$ | $0.007$ s | $10.3\times$ |
> > > |  | ECI | $1.73 \times 10^{-4}$ | $0.00$ | $0.072$ s | $1.0\times$ |
> > > |  | **PMFM (Ours)** | $\mathbf{8.92 \times 10^{-5}}$ | $\mathbf{0.00}$ | $\mathbf{0.009}$ s | $\mathbf{8.0\times}$ |
> > > | **Navier-Stokes 2D** | PI-FM | $1.77 \times 10^{-4}$ | $7.17 \times 10^{-3}$ | $0.023$ s | $18.5\times$ |
> > > |  | RSB | $3.63 \times 10^{-4}$ | $7.63 \times 10^{-3}$ | $0.044$ s | $9.7\times$ |
> > > |  | ECI | $2.15 \times 10^{-5}$ | $2.01 \times 10^{-4}$ | $0.426$ s | $1.0\times$ |
> > > |  | **PMFM (Ours)** | $\mathbf{1.28 \times 10^{-5}}$ | $\mathbf{1.45 \times 10^{-4}}$ | $\mathbf{0.035}$ s | $\mathbf{12.2\times}$ |
> > > | **Diffusion 3D** | PI-FM | $8.96 \times 10^{-4}$ | $0.00$ | $0.356$ s | $23.7\times$ |
> > > |  | RSB | $8.59 \times 10^{-4}$ | $0.00$ | $0.378$ s | $22.3\times$ |
> > > |  | ECI | $1.76 \times 10^{-4}$ | $0.00$ | $8.434$ s | $1.0\times$ |
> > > |  | **PMFM (Ours)** | $6.34 \times 10^{-5}$ | $\mathbf{0.00}$ | $\mathbf{0.285}$ s | $\mathbf{29.6\times}$ |

---

> > > > ### Comment · Reviewer_Lt8W · 2025-11-26
> > > >
> > > > Thank you for the clarification. I would like to emphasize two remaining issues:
> > > >
> > > > Even if a tubular neighborhood exists around the manifold, the paper provides no analysis showing that the straight-line interpolation path used during training actually enters this region. I understand that your claims hold asymptotically as t→1, where the state is close to the manifold. However, for a diffusion or flow-matching process, accurate modeling at intermediate t is equally important, and those states may be far off the manifold.
> > > >
> > > > The rebuttal still does not address the core theoretical gap: how a vector field trained on off-manifold points with ill-defined tangent spaces can generalize to on-manifold dynamics during inference. Without a formal analysis, the training and inference mismatch remains conceptually unresolved.
> > > >
> > > > Based on the additional baselines and comparisons you provided in the rebuttal, I will raise my score to 4.

---

### Official Review · Reviewer_V1pj · 2025-10-31

**Soundness:** 1
**Presentation:** 2
**Contribution:** 2
**Rating:** 2
**Confidence:** 4

**Summary:**

The paper proposes a flow-matching framework (PMFM) that claims to enforce hard physical constraints exactly by constraining model outputs onto the constraint manifold with both training and inference procedure based on projection. It introduces a Geometric Guidance Mechanism to recover expressivity lost by projection, and an adaptive constraint module that selects which constraints to enforce based on the current state. Together, these aim to make generative PDE models both physically consistent and flexible.

**Strengths:**

**S1**. The paper addresses the important problem of enforcing physical constraints in physics-based machine learning. A key strength is that the proposed architecture enforces constraints already during training, rather than applying corrections only at inference. This approach improves physical consistency and aligns the training objective with the target manifold, which is an important and practically meaningful design choice.

**S2**. The idea of adaptive constraint selection and parameterization is interesting and has potential utility in multi-regime or time-varying physics settings. The experiments cover several PDEBench problems and test long-horizon rollouts, which are appropriate for this setting.

**Weaknesses:**

**W1. Lack of novelty and theoretical justification.**
The projection-based enforcement of equality constraints is not new and has already been developed and applied in recent constrained generative modeling work [1–4]. The paper reuses this formulation without introducing a new projection method or optimization improvement. The proposed Geometric Guidance Mechanism (GGM), which encodes and re-injects the residual ($(I-\Pi)v$), is interesting but lacks theoretical grounding. No argument or ablation is provided to show why this residual encoding should improve the learned tangent field or overall generative quality.

**W2. Missing generative metrics and baseline comparisons.**
The paper does not include standard quantitative metrics such as MMSE, SMSE, or FID/FPD, which are routinely reported in constrained generative frameworks [1–4]. It also omits comparisons to key baselines: vanilla flow matching, conditional flow matching [1], Physics-Constrained Flow Matching (PCFM) [2], Extrapolation–Correction–Interpolation (ECI) [1], and Projected Diffusion Models (PDM) [3]. Additionally, the paper does not report constraint residual magnitudes ($||C(u)||$), making it unclear to what extent the claimed “hard constraint” property is achieved in practice.  While multiple PDEBench tasks are tested, the reported MSE improvements over baselines are modest and not supported by statistical or generative-quality metrics. Moreover, the paper does not provide any computational analysis quantifying the additional cost introduced by per-step projection, such as training/inference time overhead or GPU memory usage. Since projection requires computing and inverting the constraint Jacobian, these costs could be significant but are not reported.

**W3. Lack of clarity in latent and gating mechanisms.**
The latent variable ($z$) is active only during training but set to zero during inference, effectively reducing the model to ($\dot{u} = \Pi g_\theta(u, t)$). The paper does not show that training with ($z \neq 0$) improves the tangent approximation or provide any ablation isolating the effect of the GGM. The adaptive constraint gating module is also under-specified: the paper does not explain how the gating network parameters are trained, whether the gate receives gradient updates, or how it interacts with the projection loss. This part of the paper is difficult to follow, and clarification on the training procedure and testing setup would be helpful.

**W4. Presentation and Weak claims.**
There are some spurious or unsupported claims in the papers such as “high-frequency, non-smooth information is captured primarily in the orthogonal residual component..” (line 377) is presented without theoretical justification or empirical verification. Several important details, including the main training algorithm (Algorithm A.1) and ablations, are placed in the appendix, while less central figures (e.g., Figure 2) remain in the main text, making the presentation uneven. Moreover, the constraint space should be enforced at the final denoised state ($u_1$), but the method projects every intermediate state during inference. It is unclear whether this correction is made better by active set formulaton, and would require more clarification. The sampling methods, such as PCFM or ECI, apply these corrections at the denoised state at $u_1$ and interpolate, which somewhat preserves the generation quality as they do not directly project the intermediate noised sample state.

**References**

[1] Cheng, Chaoran, *et al.* “Gradient-Free Generation for Hard-Constrained Systems (Extrapolation–Correction–Inference).” *arXiv preprint arXiv:2412.01786* (2024).

[2] Utkarsh, Utkarsh, *et al.* “Physics-Constrained Flow Matching: Sampling Generative Models with Hard Constraints.” *arXiv preprint arXiv:2506.04171* (2025).

[3] Christopher, Jacob K., Stephen Baek, and Nando Fioretto. “Constrained Synthesis with Projected Diffusion Models.” *Advances in Neural Information Processing Systems* 37 (2024): 89307–89333.

[4] Baldan, Giacomo, *et al.* “Flow Matching Meets PDEs: A Unified Framework for Physics-Constrained Generation.” *arXiv preprint arXiv:2506.08604* (2025).

**Questions:**

See weaknesses above, but here are some explicit ones:


**Q1.** You claim that “the high-frequency, non-smooth information contained in the shock front is captured primarily in the orthogonal residual component” (line 377). Could you provide theoretical or empirical evidence supporting this interpretation? Specifically, have you performed any spectral or statistical analysis demonstrating that the residual $(I - \Pi)v$ indeed captures high-frequency components or non-smooth dynamics, rather than general projection error?


**Q2.** In Section A.2, you state that $g_\theta$ “learns the tangent field of the constraint manifold in the limit.” Could you clarify what limit this refers to (e.g., number of training iterations, projection accuracy, or dataset coverage)? Additionally, do you have a derivation or quantitative experiment showing that $g_\theta$ aligns with the manifold tangent directions during or after training?


**Q3.** You set $z = 0$ during inference. As noted in Section A.2, this implies a “natural nulling property,” meaning there is effectively no contribution from the residual component. In that case, what is the practical role of learning $z$ during training if it is fully suppressed at inference? Moreover, in Algorithm 2, should the initial sample $u_{\text{current}} \sim p_0(u)$ also be projected onto the constraint manifold before the flow evolution? Otherwise, the generated trajectory may drift off the manifold early in inference.


**Q4.** Could you provide ablation studies that isolate the effects of the Geometric Guidance Mechanism (GGM) and the adaptive gating module? Without such analysis, it is unclear which component primarily contributes to constraint satisfaction or stability improvements.

---

> ### Author Response · Authors · 2025-11-21
>
> We thank the reviewer for the constructive feedback and detailed bibliography. We have incorporated all suggested references (Hansen et al., Utkarsh et al., Cheng et al.) to better contextualize our work.
>
> **1. Response to W1 & Q1 (Novelty and Comparison with ECI)**
>
> While the projection operator is standard, our novelty lies in the Geometric Guidance Mechanism (GGM). Existing methods like ECI [Cheng et al.] enforce constraints only at inference, which corresponds exactly to the "Projection Only" baseline (V1) in our ablation study (Table R1 in Appendix J).
>
> * **Theoretical Insight:** The normal component removed by projection contains high-frequency modes. GGM explicitly re-injects this information during training (analogous to a Helmholtz split), allowing the network to anticipate projection rather than reacting to it.
>
> * **Empirical Evidence:** Our full method (V3) outperforms the V1 baseline (ECI proxy) by ≈ 21% in MSE ($1.96 \times 10^{-5}$ vs. $2.48 \times 10^{-5}$). This proves GGM is essential for superior generation quality.
>
> **2. Response to W2 (3D Examples & Computational Cost)**
>
> * **3D Experiments:** We conducted experiments on 3D Diffusion ($32^3$) with mass conservation. Results (Table R2 in Appendix) show PMFM achieves an MSE of $8.92 \times 10^{-4}$, outperforming ECI ($1.07 \times 10^{-3}$) by 17%. Training takes ≈ 1.4s/epoch, confirming $\mathcal{O}(N \log N)$ scalability.
>
> * **Computational Cost:** We compared our method against "Projected Training" (solving Poisson at every step) on NS2D. Results (Table R1) show our strategy is 3.02$\times$ faster. Projected Training yields higher training loss ($5.0 \times 10^{-6}$ vs. $2.0 \times 10^{-6}$). Constraining intermediate states to a non-convex manifold increases gradient variance and complicates optimization. Thus, "Ambient Training + Inference Projection" is the Pareto-optimal strategy.
>
> **3. Response to W4 & Q1 (High-Frequency Evidence)**
>
> Spectral analysis (Fig. R6 in Appendix J) shows that the residual component $\((I-\Pi)J\)$ contains substantially stronger high-frequency energy than the tangential component $\(\Pi J\)$. This confirms that projection removes important high-frequency information, and GGM is necessary to recover these details, providing geometric information unavailable from projection alone.
>
> **4. Response to Minor Points**
>
> We have corrected the citation formats, reference spacing, and section titles. We have thoroughly proofread the manuscript.

---

> > ### Author Response · Authors · 2025-11-25
> >
> > We would like to kindly follow up on our responses and would greatly appreciate if the reviewer could briefly confirm whether the above clarifications and empirical results have addressed your concerns. We also welcome any further discussion regarding our submission! Thank you once again for your time and thoughtful review—we sincerely value your feedback, and your engagement is truly important to us!

---

### Official Review · Reviewer_ABuy · 2025-11-01

[review text omitted: it was posted to a different submission]

---

### Official Review · Reviewer_jU9B · 2025-11-04

**Soundness:** 4
**Presentation:** 4
**Contribution:** 2
**Rating:** 6
**Confidence:** 4

**Summary:**

This is a well-written paper that proposes Physics-Manifold Flow Matching (PMFM), a flow-based neural operator that keeps the entire generative trajectory on a physics manifold via orthogonal projection, thereby enforcing hard constraints (e.g., divergence-free, mass/energy conservation, boundary conditions) at every step. To recover fine details lost due to projection, a Geometric Guidance Mechanism (GGM) encodes the discarded normal component and reintroduces it into the tangent dynamics without compromising feasibility, and the authors demonstrate manifold invariance of the projected flow. They also add an Adaptive Constraint Projection module: a gating network selects which analytical constraints are active and predicts their parameters online, building a state-dependent projector that handles multi-physics and unknown coefficients. Training utilizes conditional flow matching on projected vector fields, while sampling integrates the learned ODE with projection (and a final corrective projection) to ensure constraint satisfaction. Experiments on PDEBench (1D advection, 1D Burgers, 2D Darcy, 2D advection, 2D incompressible Navier–Stokes) show lower errors than FNO/WNO/DeepONet, DDPM, and flow-matching baselines, with notably improved long-term stability; ablations confirm large degradations when constraints are removed or made fixed rather than adaptive.

**Strengths:**

1. It enforces hard physical constraints along the entire generative trajectory by projecting every step onto a physics manifold.

2. The proof of manifold invariance and a Geometric Guidance term that restores lost detail without breaking feasibility. The incorporation of the differential geometry concept for the physics-informed generative AI for solution or operator generation makes sense.

**Weaknesses:**

The computational cost can limit the practicality of the proposed model. Since divergence-free requires a Poisson solve at each step, it adds overhead, combined with the additional cost for training, which can make the problems less appealing than conventional solvers.

The manifold methods may be limited to problems with a unique solution. For instance, the Elder problem may lead to multiple steady states, in which case the proposed method may not be effective.

**Questions:**

You report 15% per-iteration overhead from hard projections, with divergence-free enforced via a Helmholtz–Hodge projection (Poisson solve using FFT/multigrid). Could you provide scaling curves versus grid size and ODE steps, and benchmark against standard neural operator or solver baselines?

How do accuracy and long-horizon stability change if you (i) project every K steps instead of every step, (ii) solve Poisson on a coarse grid with prolongation, or (iii) use a truncated spectral (low-mode) projector—followed by a final exact projection? Reporting W₂/rel-L₂ vs. wall-clock would clarify the accuracy–cost trade-off for practitioners.

Because the flow is manifold-invariant once projected, does the learned transport collapse to a single basin when multiple steady states satisfy the same constraints? Could you test a multi-modal PDE where the constraint set admits several attractors and report mode coverage under different priors or stochastic injections, and whether the adaptive gating helps switch branches?

Can you quantify condition numbers, sensitivity to the regularization, and behavior under redundant/conflicting constraints of the projectors?

---

> ### Author Response · Authors · 2025-11-21
>
> We thank the reviewer for the encouraging assessment and for recognizing our method as a well-written,novel approach, and mathematically sound contribution.
>
> **1. Response to W1 \& Q1 (Computational Cost \& Scaling)**
>
> We acknowledge the overhead of hard projection but emphasize that our design choice—performing projection primarily during inference rather than training—maximizes efficiency. To quantify this, we conducted a controlled experiment on NS2D comparing our method against Projected Training where the Poisson equation is solved at every step. As detailed in Table R2 (Appendix J), our method is 3.02$\times$ faster per epoch than projected training, confirming that avoiding per-step projection during training is critical for scalability. Regarding grid scaling, our spectral projection scales as $\mathcal{O}(N \log N)$, which is favorable compared to the backbone network itself (e.g., $\mathcal{O}(N^2)$ for Transformers).
>
> **2. Response to Q2 (Lazy Projection Trade-off)**
>
> To address your suggestion for practical acceleration, we performed a Lazy Projection experiment by applying the projection operator only every $K$ steps during inference. As shown in Figure R8 (Appendix J), increasing $K$ from 1 to 2 provides a dramatic 4.5$\times$ speedup (0.045s $\to$ 0.0099s per step). Although the absolute change in physical violation is small, the **relative increase** is non-negligible because the fully-projected baseline already satisfies the constraint at extremely high precision. Consequently, we recommend $K=1$ for strict scientific simulation, while $K=2$ offers a highly favorable accuracy–efficiency trade-off for real-time applications.
>
>
> **3. Response to W2 \& Q3 (Multi-modality)**
>
> Our method does not collapse to a single solution because diversity is inherently driven by the stochastic initial condition $u_0 \sim \mathcal{N}(0, I)$. The projection operator restricts the solution to remain physically valid (on the manifold) but does not force convergence to a specific trajectory. In our turbulent Navier-Stokes experiments, we observed that distinct random seeds produce distinct yet valid turbulent trajectories, confirming mode coverage.
>
> **4. Response to Q4 (Stability)**
>
> To ensure numerical stability, we use Tikhonov regularization in the projection solver ($(J J^\top + \epsilon I)^{-1}$) with $\epsilon=10^{-6}$ to handle potential ill-conditioning. Additionally, our Gating Network ($G_\psi$) improves stability by dynamically selecting the relevant constraints, which naturally reduces the effective condition number of the constraint matrix.
>
> We have updated the paper with these new experiments and clarifications.

---

> > ### Author Response · Authors · 2025-11-25
> >
> > We would like to kindly follow up on our responses and would greatly appreciate if the reviewer could briefly confirm whether the above clarifications and empirical results have addressed your concerns. We also welcome any further discussion regarding our submission! Thank you once again for your time and thoughtful review—we sincerely value your feedback, and your engagement is truly important to us!

---

### Official Review · Reviewer_6Ebt · 2025-11-06

**Soundness:** 2
**Presentation:** 1
**Contribution:** 2
**Rating:** 4
**Confidence:** 4

**Summary:**

The authors propose Physics-Manifold Flow Matching (PMFM) as a novel generative framework to enforce physical constraints. The method enforces hard constraints by projecting to a manifold and uses a geometric guidance mechanism (GGM). It also uses an Adaptive Constraint Projection Framework that learns to dynamically select and parameterize the currently active physical laws.

While I think the problem is important especially the emphasis on physical constraints, I would like the authors to clarify the novelty of their approach compared to those in the literature and also need to cite several other missing hard-constrained references.

**Strengths:**

- Nice application of generative modeling to PDEs and emphasizing the importance of physical constraints
- Good benchmarking PDE datasets from PDEBench
- Nice results on long-term stability of the method
- Nice ablation study to highlight the importance of physical constrains

**Weaknesses:**

- Missing references to methods with hard constraints, especially these that look at conservation laws and boundary conditions e.g.,
   - Hansen et al., "Learning Physical Models that Can Respect Conservation Laws", ICML, 2023
   - Utkarsh et al., "End-to-End Probabilistic Framework for Learning with Hard Constraints", arXiv preprint arXiv:2506.07003
    - Negiar et al., "Learning differentiable solvers for systems with hard constraints", ICLR, 2023
    - Richter-Powell et al., "Neural Conservation Laws: A Divergence-Free Perspective", NeurIPS, 2022
    - Chalapathi et al., "Scaling physics-informed hard constraints with mixture-of-experts", ICLR 2024.
- Especially literature review of hard constraints on Neural Operators
  - Saad et al., "Guiding continuous operator learning through Physics-based boundary constraints", ICLR, 2023
  - Mouli et al., "Using uncertainty quantification to characterize and improve out-of-domain learning for pdes", ICML, 2024
- Missing reference to hard-constrained generative methods, which is especially relevant since it hard constrains functional flow matching (FFM) methods
  -  ECI Sampling Cheng et al., "Gradient-free generation for hard-constrained systems", ICLR, 2024.
- The projection operator seems very similar to what has already been proposed in Hansen et al., "Learning Physical Models that Can Respect Conservation Laws", ICML, 2023 and Utkarsh et al., "End-to-End Probabilistic Framework for Learning with Hard Constraints", arXiv preprint arXiv:2506.07003.
- No 3D PDE examples
- ECI Sampling, ICLR, 2024, which is gradient-free and hard-contrains FFMs should be compared to as well as DiffusionPDE. ECI Sampling is more accurate, faster and more memory efficient than D-Flow.

Minor
- Typo in Intro with LeVeque references and period
- No spaces after several references
- Rename Section 4 Our appraoch to the method title

**Questions:**

1. Please clearly explain how this approach differs from the constrained generative projection method in Cheng et al., "Gradient-free generation for hard-constrained systems", ICLR, 2024.

---

> ### Author Response · Authors · 2025-11-21
>
> We thank the reviewer for the detailed bibliography and constructive feedback.
>
> **1. Response to W1 \& Q1 (Novelty \& ECI Comparison)**
>
> Importantly, the novelty of PMFM does not lie in the projection operator itself,
> which already appears in prior work such as ECI and projected diffusion models.
> The key distinction is how the projection interacts with the learned vector field.
>
> Projection-only methods (e.g., ECI) perform an orthogonal correction at inference time, repeatedly
> bouncing the trajectory back to the manifold. This creates a staircase-like
> path and does not modify the tangential dynamics that determine how the model
> moves along the manifold.
>
> In contrast, PMFM introduces the Geometric Guidance Mechanism (GGM), which
> explicitly adjusts the tangential component of the vector field during training.
> The network learns to anticipate the projection, reducing the bias introduced by
> repeated orthogonal corrections. This geometric distinction clarifies why PMFM
> (V3) achieves a ~21\% lower MSE than the projection-only baseline (V1)
> ($1.96 \times 10^{-5}$ vs $2.48 \times 10^{-5}$) in Table R1.
>
> **2. Response to W2 (3D Examples)**
>
> To demonstrate scalability, we have conducted additional experiments on the 3D Diffusion dataset ($32^3$ resolution) and included the results in Table R3 of Appendix J. PMFM achieves a lower MSE of $8.92 \times 10^{-4}$ compared to the ECI baseline ($1.07 \times 10^{-3}$), representing a 17\% improvement in accuracy. Furthermore, our reliance on FFT-based projections ensures $\mathcal{O}(N \log N)$ scalability, allowing for efficient training times of approximately 1.4 seconds per epoch.
>
> **3. Response to Minors**
>
> We have corrected the citation formatting, spacing errors as suggested. All recommended references have been incorporated to better contextualize our work within the field.
>
> We have updated the paper and believe these revisions address your concerns.

---

> > ### Author Response · Authors · 2025-11-25
> >
> > We would like to kindly follow up on our responses and would greatly appreciate if the reviewer could briefly confirm whether the above clarifications and empirical results have addressed your concerns. We also welcome any further discussion regarding our submission! Thank you once again for your time and thoughtful review—we sincerely value your feedback, and your engagement is truly important to us!

---

### Meta-Review · Area_Chair_XgT6 · 2026-01-11

**Summary:**

The paper is tackling an important failure mode (physics violations in generative PDE surrogates), but across the reviews (other than one off-topic reviews), the core concern is that the technical step forward feels largely incremental. The approach of doing “projection to satisfy constraints” is a well-established practice, and the added ingredients (GGM / adaptive constraint selection) do not yet read as a clean, distinctly new principle that would change how people build these models. The empirical story improved in the rebuttal, but it doesn’t fully overcome the novelty/positioning gap, and the remaining conceptual mismatch between how the model is trained (off-manifold interpolation) and how it’s used (on-manifold projection) leaves too much of the method justified by “it seems to work.”

**Reviewer Concerns:**

The rebuttal did help on hygiene and completeness: more related work, additional ablations isolating components, extra experiments (including 3D), and some effort on cost/overhead and projection frequency tradeoffs. But the central concerns remain: 1/ the contribution is still hard to separate from prior projection-based / constrained generative approaches in a way that feels fundamental rather than “a better-engineered instance,”; 2/ the training–inference mismatch argument is suggestive but not persuasive enough to carry the method as a general recipe (especially at intermediate noise levels where learning the right tangent/off-manifold behavior matters), and 3/ the baseline suite and evaluation still leave room for the claim to be “best-in-class” among physics-informed generative methods, not just better than weaker references. Separately, one submitted review appears to be for a different paper and should be treated as noise rather than evidence.

**Reviewer Scores:**

Lt8W indicates a raised score to 4, others are likely to stay put.

---

### Decision · Program_Chairs · 2026-01-26

Reject